

# Species interactions can shift the response of a maerl bed community to ocean acidification and warming

Erwann LEGRAND[1,2], Pascal RIERA[1,2], Mathieu LUTIER[1,2], Jérôme COUDRET[1,2], Jacques GRALL[3], Sophie MARTIN[1,2]

[1]Sorbonne Universités, UPMC Univ Paris 06, Station Biologique, Place Georges Teissier, 29688 Roscoff Cedex, France
[2]CNRS, Station Biologique, Place Georges Teissier, 29688 Roscoff Cedex, France
[3]UBO, IUEM, Place Nicolas Copernic, 29280 Plouzané, France

*Correspondence to*: Erwann Legrand (erwann.legrand@sb-roscoff.fr)

## Abstract

Predicted ocean acidification and warming are likely to have major implications for marine organisms, especially marine calcifiers. However, little information is available on the response of marine communities as a whole to predicted changes. Here, we experimentally examined the combined effects of temperature and partial pressure of carbon dioxide ($pCO_2$) increases on the response of maerl bed assemblages, composed of living and dead thalli of the free-living coralline alga *Lithothamnion corallioides*, epiphytic fleshy algae, and grazer species. Two three-month experiments were performed in the

winter and summer seasons in mesocosms with four different combinations of $pCO_2$ (ambient and high $pCO_2$) and temperature (ambient and + 3°C). The response of maerl assemblages was assessed using metabolic measurements at the species and assemblage scales. Gross primary production and respiration of assemblages were enhanced by high $pCO_2$ conditions in the summer. This positive effect was attributed to the increase in epiphyte biomass, which benefited from higher $CO_2$ concentrations for growth and primary production. Conversely, high $pCO_2$ drastically decreased the calcification

rates in assemblages. This response can be attributed to the decline in calcification rates of living *L. corallioides* due to acidification as well as increased dissolution of dead *L. corallioides*. Future changes in $pCO_2$ and temperature are likely to promote the development of non-calcifying algae to the detriment of the engineer species *L. corallioides*. The development of fleshy algae may be modulated by the ability of grazers to regulate epiphyte growth. However, our results suggest that predicted changes will negatively affect the metabolism of grazers and potentially their ability to control epiphyte

abundance. Here, we demonstrate that the response of marine communities to climate change will depend on the direct effects on species physiology and the indirect effects due to shifts in species interactions. This double, interdependent



response underlines the importance of examining community-level processes, which integrate species interactions, to better understand the impact of global change on marine ecosystems.

## 1.    Introduction

A growing body of literature predicts that ocean acidification and warming will be the main anthropogenic drivers affecting marine species by the end of the century (Kroeker et al., 2013). Due to the increase in atmospheric $CO_2$, seawater surface temperatures have been predicted to increase by 0.71-2.73°C and pH to decline by 0.07-0.33 units in the surface ocean by the end of the 21[st] century (Bopp et al., 2013).

Species interactions are a key element in ecosystem functioning and are likely to play an important role in species responses

to climate change (O'Connor et al., 2011; Hansson et al., 2012; Kroeker et al., 2012). To date, most research has focused on the impact of ocean acidification and warming on the response of single species (Yang et al., 2016); studies examining the effects of climate change on marine communities are scarce in the literature (Alsterberg et al., 2013). Understanding the mechanisms and interactions that occur among marine communities that face the predicted changes is necessary for a better overview of marine ecosystem response. Climate change is likely to strongly alter interactions between macroalgae (e.g.

calcifying and non-calcifying macroalgae; Olabarria et al., 2013; Short et al., 2014; Short et al., 2015), interactions between grazers and macroalgae (Poore et al., 2016; Sampaio et al., 2017) as well as prey-predator dynamics (Asnaghi et al., 2013; Jellison et al., 2016), inducing drastic consequences on the structure and functioning of marine ecosystems (Widdicombe and Spicer, 2008; Hale et al., 2011).

Maerl beds feature high structural and functional diversity arising primarily from the numerous species interactions that

occur in this environment — in particular, interactions between fleshy and calcareous macroalgae and grazers and macroalgae (Hily et al., 1992; Guillou et al., 2002; Grall et al., 2006). The accumulation of living and dead thalli of free-living coralline algae (Corallinaceae, Rhodophyta) creates a complex three-dimensional structure that provides habitat for many faunal and floral species (Foster et al., 2007; Amado-Filho et al., 2010; Peña et al., 2014), some of which have high commercial value (Grall and Hall-Spencer, 2003). In some locations, dead maerl can reach high proportions compared with

living maerl (Hily et al., 1992), thereby contributing substantially to the local carbonate dynamics (Martin et al., 2007).

The main species inhabiting maerl beds may respond differently to ocean acidification and warming. Coralline algae are known to be among the most vulnerable species facing ocean acidification (McCoy and Kamenos, 2015; Martin and Hall-Spencer, 2016), due to their highly soluble Mg-calcite skeleton (Morse et al., 2006). The deleterious consequences of ocean acidification have also been demonstrated for other calcareous marine taxa, such as mollusks (Gazeau et al., 2013; Parker et al., 2013) and echinoderms (Dupont et al., 2010), with reductions in survival, growth, development, and abundance (Kroeker et al., 2013). Conversely, some species can benefit from the increase in $CO_2$ concentration and temperature. Positive responses, such as increases in primary production and growth, have been found mostly among non-calcifying organisms, such as fleshy algae and seagrasses (Koch et al., 2013; Pajusalu et al., 2013).

Here, we experimentally investigated the impact of ocean acidification and warming on the metabolism and the interactions of the main maerl-forming species in Brittany *Lithothamnion corralloides* and the epiphytic fleshy macroalgae and main grazer (gastropods and sea urchins) associated with it. Because the response of species and communities to climate change is also likely to vary depending on seasonal changes in environmental factors (Godbold and Solan, 2013; Martin et al., 2013; Baggini et al., 2014), experiments were performed in both winter and summer conditions. The response of marine communities to climate change is likely to be influenced by the direct effects of environmental stressors on individual organisms, and by the indirect effects induced by shifts in interspecific interactions (Harley et al., 2012; Auster et al., 2013). In the present study, we therefore performed metabolic measurements at the species and at the community scale. At the species scale, studying species physiology is useful for understanding how organisms cope with changing climatic conditions and for analyzing the community metabolic response. Community-scale measurements provide information on the potential shifts in species interactions induced by climate change. In particular, we tested the hypothesis that climate change will increase epiphytic fleshy algal growth, exacerbating the deleterious consequences of predicted changes on *L. corallioides* metabolism. We also investigated whether the predicted changes can modify interactions between grazers and macroalgae, and their ability to regulate epiphytic biomass.



## 2.     Materials and methods

### 2.1.     Species collection and assemblages

75  Organisms were collected from a maerl bed in the Bay of Brest, France (48°18'N 4°23'W) using a naturalist's dredge (width:

1 m, height: 0.2 m, net: 1.5 m long) deployed from the research vessel *Albert Lucas*. In the Bay of Brest, maerl beds are

located at depths of between 0.7 and 6.8 m, according to the tide (Dutertre et al., 2015). We deliberately selected thalli of the

maerl species *L. corallioides* Crouan and Crouan, 1867 that were devoid of any apparent epiphytes; nonetheless, they were

not cleaned so as to retain any epiphyte spores that may have been present on their surface. Medium-sized individuals of the

three main species of grazers living in maerl beds were also sampled: two gastropod species (sea snails) *Gibbula magus*

Linnaeus, 1758 and *Jujubinus exasperatus* Pennant, 1777 and an urchin species *Psammechinus miliaris* Müller, 1771 (Grall

et al., 2006). Samples were collected on 24 January 2015 (winter conditions) and 15 September 2015 (summer conditions).

In each season, 1 kg of living thalli of *L. corallioides*, 500 g of dead thalli of *L. corallioides*, 40 individuals of *G. magus*

(shell length range 17-29 mm; Table S1), 40 individuals of *P. miliaris* (test diameter range 11-23 mm), and 80 individuals of

*J. exasperatus* (shell height range 5-11 mm) were randomly selected and transported in seawater tanks to the Roscoff Marine

Station. To mitigate the stress experienced by the species during sampling and transport, they were kept in open-flow aquaria

at ambient pH and *in situ* temperature conditions at the time of collection for at least one week before starting the

experiments. No mortality was recorded during this period.

### 2.2.     Experimental design

Two three-month long experiments were conducted for both winter (March to June 2015) and summer (September to

December 2015) conditions.

For each season, 20 artificial assemblages were created and randomly assigned to 20 15 L aquaria. Each assemblage was

composed of 45 g of living *L. corallioides* thalli, 20 g dead *L. corallioides* thalli, two *G. magus* individuals, two *P. miliaris*

individuals and four *J. exasperatus* individuals, according to the proportions observed on maerl beds.

Algae and grazers were acclimated to laboratory conditions for 7 days. Then, the pH was gradually decreased by 0.05 units

per day over 7 days and temperature increased by 0.5°C per day. The pH was controlled by modifying $pCO_2$ through $CO_2$

bubbling.





At each season, two $pCO_2$ conditions were tested, each with two temperature conditions to examine the interaction between

$pCO_2$ and temperature. There were therefore four conditions:

1) ambient $pCO_2$ and ambient temperature (control, A-$pCO_2$; T)

2) high $pCO_2$ and ambient temperature (H-$pCO_2$; T)

3) ambient $pCO_2$ and high temperature (A-$pCO_2$; T + 3°C)

4) high $pCO_2$ and high temperature (H-$pCO_2$; T + 3°C).

Ambient $pCO_2$ conditions (A-$pCO_2$) were determined according to *in situ* winter (7.98) and summer (8.06) mean $pH_T$ (pH on

the total scale) monitored above maerl beds in the Bay of Brest (data from Martin, unpublished data). High $pCO_2$ (H-$pCO_2$)

corresponded to the "business-as-usual" scenario predicted for the end of the century, with a pH decrease of -0.33 units

(RCP8.5; Bopp et al., 2013). Ambient temperature (T) corresponded to *in situ* winter (10.0°C) and summer (17.1°C)

conditions in the Bay of Brest recorded by SOMLIT (from 2003 to 2014), and high temperature (T + 3°C) was determined

according to the business-as-usual scenario predicted for 2100 (Bopp et al., 2013).

The pH and the temperature were controlled in four 100 L tanks, continuously supplied with filtered (5 µm) natural seawater,

with a high water flow rate of 150 L h$^{-1}$ per tank. They were maintained by an off-line feedback system (IKS Aquastar,

Karlsbad, Germany) that activated or stopped heaters and solenoid valves, controlling temperature and $CO_2$ (Air Liquide,

France) bubbling in the tanks, respectively. Each 100 L tank provided seawater to five 15 L aquaria for each of the four

conditions using pumps. The water flow rate was 15 L h$^{-1}$ in each aquarium. Temperature was maintained constant in aquaria

with water baths. Seawater pH ($pH_T$, expressed on the total hydrogen ion concentration scale, Dickson et al., 2007) and

temperature were monitored every two days in the 20 aquaria, at different times of the day. Seawater $pH_T$ and temperature

measurements were carried out using a pH probe associated with a temperature sensor (PHC101, Hach Lange, IntelliCAL).

The pH probe was calibrated using Tris/HCl and 2-aminopyridine/HCl buffers (Dickson et al., 2007). The pH values of the

off-line feedback system were adjusted from measurements of $pH_T$ carried out every two days in each aquarium. Total

alkalinity ($A_T$) was also monitored during the experiment in each aquarium at different times of the day (n = 28). For $A_T$

analyses, seawater samples (60 mL) were filtered through 0.7 µm Whatman GF/F filters and immediately poisoned with a

mercuric chloride solution to prevent further biological activity (Dickson et al., 2007). $A_T$ was determined using open-cell

titration on an automatic titrator (Titroline alpha, Schott SI Analytics, Mainz, Germany) according to the method developed

by Dickson et al. (2007). $A_T$ was calculated using a Gran function applied to pH values ranging from 3.5 to 3.0 (Dickson et

al., 2007) and corrected using standard reference material provided by the Andrew G. Dickson laboratory (CRM Batch 111,

accuracy of $\pm$ 6 µmol kg$^{-1}$). Salinity was measured every 2 weeks with a conductivity probe (CDC401, Hach Lange,

IntelliCAL, accuracy of 0.1) and remained constant during experiments (35.2 $\pm$ 0.2). From $A_T$ and $pH_T$ measurements,

dissolved inorganic carbon (DIC), saturation state of seawater with respect to aragonite ($\Omega_{Ar}$) and saturation state of seawater

with respect to calcite ($\Omega_{Ca}$) were calculated with CO2SYS software. Mean temperature and parameters of the carbonate

chemistry are given in Table 1.

Irradiance was set to the mean *in situ* daily irradiance at 5 m depth in the Bay of Brest according to Martin et al. (2006): 30-

40 µmol photons m$^{-2}$ s$^{-1}$ in winter and 90-100 µmol photons m$^{-2}$ s$^{-1}$ in summer. The light was provided by two or four 80 W

fluorescent tubes (JBL Solar Ultra Marin Day, JBL Aquaria, Nelson, New Zealand) above the aquaria under a 10/14 h or

14/10 h light/dark photoperiod, for winter or summer conditions, respectively.

**2.3.    Metabolic measurements**

After three months in experimental conditions, metabolic measurements were performed at the species and assemblage level

using incubations in acrylic respirometry chambers (Engineering and Design Plastics Ltd, Cambridge, UK). For species-

scale measurements, each species was incubated separately. Community-scale measurements were performed on

assemblages, incubating all individuals from all species present in each aquarium. The chamber volume was adapted to

species size. It was of 80 mL for *J. exasperatus* and epiphytes, 185 mL for *P. miliaris, G. magus* and living and dead *L.

corallioides*, and 600 mL for the assemblages. Before incubation, epiphytic algae that spontaneously grew on *L. corallioides*

during the experiments were carefully removed and incubated separately. Metabolic measurements (net photosynthetic and

respiration rates) for the main epiphytic algae *Rhodymenia ardissonei* and *Solieria chordalis* were only examined in the

summer, when their biomass was sufficient for measurements. Species were placed on a plastic grid above a stir bar in the

chambers to ensure the seawater was well mixed. For *G. magus* and *P. miliaris*, net calcification, respiration and excretion

(ammonia release) rates were measured. For *J. exasperatus*, only respiration rates were measured due to its limited size and

metabolic rates. For grazers, physiological rates were measured under ambient irradiance. For each grazer species,





individuals present in each aquarium were incubated together. For living and dead *L. corallioides* and assemblages, net

photosynthetic and light calcification rates were measured under ambient irradiance, and respiration and dark calcification

rates were measured in the dark. For light incubations, chambers were placed inside aquaria to control temperature. For dark

incubations, chambers were placed in a plastic crate filled with aquaria seawater in an open circuit to keep the temperature

constant. Incubation duration was adjusted to keep oxygen saturation above 80%. Incubations lasted approximately from 1 h

for *G. magus* to 2.5 h for dead maerl. For assemblages, the metabolism was measured from the incubations of all species

together.

Oxygen concentrations were measured at the beginning and at the end of each incubation, using an optical fiber system

(FIBOX 3, PreSens, Regensburg, Germany). Reactive spots were calibrated with 0% and 100% buffer solutions. Net primary

production (NPP, $\mu$mol $O_2$ g $DW^{-1}$ $h^{-1}$) or respiration (R, $\mu$mol $O_2$ g $DW^{-1}$ $h^{-1}$) rates were calculated following Eq. (1):

$$\text{NPP or R} = \frac{\Delta O_2 \times V}{\Delta t \times DW} \quad (1)$$

where $\Delta O_2$ is the difference between the initial and final oxygen concentrations ($\mu$mol $O_2$ $L^{-1}$), V the volume of the chamber

(L), $\Delta t$ the incubation time (h), and DW the dry weight of the species incubated (g). The dry weight was obtained after 48 h

at 60°C. For gastropods, the body was separated from the shell to consider the dry weight of the body only.

For algae and the assemblages, gross primary production (GPP) was calculated following Eq. (2):

$$\text{GPP} = \text{NPP} - \text{R} \quad (2)$$

Control incubations containing only seawater were carried out to correct for oxygen fluxes due to any additional biological

activity in seawater. Oxygen fluxes calculated in control chambers were subtracted from oxygen fluxes of chambers

containing algae.

Seawater samples were taken in the aquaria at the beginning of the incubation and in the chambers at the end of the

incubations (except for fleshy algae and *J. exasperatus*) to measure ammonium ($NH_4^+$) concentration and total alkalinity

($A_T$). To do so, 45 mL seawater samples for $NH_4^+$ analyses were fixed with reagent solutions and stored in the dark. $NH_4^+$

concentrations were determined according to the Solorzano method (Solorzano, 1969). Absorbance was measured by

spectrophotometry at a wavelength of 630 nm (spectrophotometer UV-1201V, Shimadzu Corp, Kyoto, Japan). For grazers,

ammonia excretion rates (E, $\mu$mol $NH_4^+$ g $DW^{-1}$ $h^{-1}$) were calculated following Eq. (3):



$$E = \frac{\Delta NH_4^+ \times V}{\Delta t \times DW} \quad (3)$$

where $\Delta NH_4^+$ is the difference between the initial and final ammonium concentrations ($\mu mol\ NH_4^+\ g\ DW^{-1}\ h^{-1}$).

For $A_T$ analyses, 60 mL seawater samples were filtered through 0.7 µm Whatman GF/F filters and were immediately poisoned with a mercuric chloride solution. Total alkalinity was determined according to the method described above. Net

calcification rates at light and in the dark ($G_l$ and $G_d$, respectively; in $\mu mol\ CaCO_3\ g\ DW^{-1}\ h^{-1}$) were calculated according to the alkalinity anomaly technique (Smith and Key, 1975) and corrected for $NH_4^+$ fluxes (Gazeau et al., 2015). This correction was applied to calcareous species and assemblage incubations following Eq. (4):

$$G_l\ or\ G_d = \frac{(-\Delta A_T + \Delta NH_4^+) \times V}{2 \times \Delta t \times DW} \quad (4)$$

where $G_l$ is the net calcification in the light, $G_d$ is the net calcification in the dark, $\Delta A_T$ is the difference between the initial and final $A_T$ ($\mu eq\ L^{-1}$).

After the three-month experiments, epiphytic algae that spontaneously grew on *L. corallioides* during experiments were picked off and dried at 60°C for 48 h to determine their dry weight.

## 2.4.    Chlorophyll *a* analysis

At the end of the experiments, thalli of living and dead *L. corallioides* were collected in each aquarium and immediately frozen at -20°C pending analyses. Then samples were freeze-dried and crushed into a powder using a mortar, in the dark. An

aliquot of 0.15 g of powder was precisely weighed and suspended in 10 mL of 90% acetone and stored in the dark at 4°C for 12 h. Samples were then centrifuged at 4000 rpm. The supernatant was collected and absorbance was measured at 630 ($A_{630}$), 647 ($A_{647}$), 664 ($A_{664}$), and 691 ($A_{691}$) nm. Chlorophyll *a* (Chl *a*) concentrations ($\mu g\ g\ DW^{-1}$) were calculated from Ritchie (2008) following Eq. (5):

$$Chl\ a = \frac{(-0.3319\ A_{630} - 1.7485\ A_{647} + 11.9442\ A_{664} - 1.4306\ A_{691}) \times V}{mp} \quad (5)$$

where V is the volume of acetone (mL) and *mp* the mass of powder (g).

**2.5.     Data analysis**

The influence of season, temperature and $pCO_2$ was tested on metabolic rates of grazers (*P. miliaris*, *G. magus* and *J. exasperatus*), living and dead maerl, epiphytic biomass and assemblages. Even after transformations, the data were non-normally distributed. Therefore, analyses were conducted using a three-way permutational multivariate analysis of variance (PERMANOVA), based on Euclidian distance (Anderson, 2001). PERMANOVAs were run with 4999 permutations

(Anderson, 2001), using season (two levels: winter and summer), temperature (two levels: ambient and elevated temperature) and $pCO_2$ (two levels: ambient and elevated $pCO_2$) as fixed orthogonal factors (n = 5). These statistical analyses were performed with the PRIMER 7 & PERMANOVA+ software package.

The effects of $pCO_2$ and temperature on the physiological rates of the epiphytic algae *R. ardissonei* and *S. chodalis* were only tested in the summer. Because assumptions of normality (Shapiro test) and homogeneity of variances (Bartlett test)

were not met, two-way non-parametric Scheirer-Ray-Hare tests were performed. These statistical analyses were carried out using the statistical package R, version 3.2.2.

**3.     Results**

**3.1.     Metabolic responses of grazers to acidification and warming**

In the urchin *P. miliaris*, high temperature (+3°C) reduced *P. miliaris* R in the summer, while $pCO_2$ had no significant effect

on *P. miliaris* R (Fig. 1a; Table 2). *P. miliaris* $G_1$ was significantly affected by the triple interaction between season, temperature and $pCO_2$ (Fig. 1b), with a negative impact due to the combined effect of temperature and $pCO_2$ increase in the summer, but no interaction effects in the winter. The combined increase of temperature and $pCO_2$ significantly affected *P. miliaris* E (Table 2; Fig. 1c).

Neither temperature nor $pCO_2$ increases significantly affected *G. magus* R, $G_1$ and E (Table 2; Fig. 1d-f). In *J. exasperatus*, R

was positively affected by the temperature increase, but in winter conditions only (Table 2; Fig. 1g). *J. exasperatus* R was negatively influenced by the $pCO_2$ increase in the winter, but positively in the summer.



### 3.2.    Metabolic responses of living *L. corallioides* to acidification and warming

Living maerl GPP did not differ among temperature and pCO$_2$ conditions regardless of the season (Table 3; Fig. 2b,c). R was significantly reduced by the high temperature condition in the winter, whereas an increase in R was observed in the summer.

Chlorophyll *a* content was negatively affected by the high temperature condition in the winter only (Tables 3,4). Temperature had a positive effect on the G$_l$ of living maerl. Conversely, G$_l$ was significantly reduced under high pCO$_2$ (Table 3; Fig. 2d). G$_d$ was significantly affected by an interaction between season and temperature (Fig. 2e). G$_d$ was positively affected by temperature in winter, but no effect was detected in the summer. A significant decline in G$_d$ occurred under high pCO$_2$ regardless of the season. Net dissolution, because G$_d$ was negative, was recorded in the winter under high

pCO$_2$ conditions.

### 3.3.    Metabolic responses of dead *L. corallioides* to acidification and warming

The high temperature condition (+3°C) did not affect dead maerl GPP or R (Table 3; Fig. 2g,h). The pCO$_2$ increase did not affect dead maerl GPP in either season. However, there was an interaction between season and pCO$_2$, with a decrease in R under high pCO$_2$ in the summer. Chlorophyll *a* content was significantly affected by the temperature and pCO$_2$ interaction

(Tables 3,4). Dead maerl G$_l$ significantly increased under high temperature (Fig. 2i). Conversely, a negative impact of high pCO$_2$ was on G$_l$ in the winter and summer. In the dark, net dissolution was observed on dead maerl regardless of the temperature and pCO$_2$ conditions (Fig. 2j). No temperature effect was observed on dark dissolution. However, dark dissolution rates were significantly higher under high pCO$_2$ treatments, regardless of the season.

### 3.4.    Growth and metabolic responses of epiphytic algae to acidification and warming

Mean GPP and R for the two epiphytic algae *R. ardissonei* and *S. chordalis* measured in the summer are presented in Figure 3. *R. ardissonei* GPP was not affected by high temperature or pCO$_2$ conditions, and R was reduced under high pCO$_2$ (Table 5; Fig. 3b,c). In *S. chordalis*, GPP was significantly affected by the interaction between temperature and pCO$_2$ (Table 5; Fig. 3e). R was enhanced by the high temperature and pCO$_2$ conditions (Fig. 3f).

The mean biomass of epiphytic fleshy algae at the end of the experiment was significantly higher in the summer than in the

winter (+81%, 3-way PERMANOVA, df = 1, F = 5.3, p=0.027, Fig. 4). Epiphyte biomass was significantly affected by the

triple interaction between season, temperature and $pCO_2$ (3-way PERMANOVA, df = 1, F = 4.9, p=0.035), with high $pCO_2$ having a positive effect on epiphyte biomass in the winter. In the summer, this positive effect was only detected under high temperature.

### 3.5.    Metabolic responses of assemblages to acidification and warming

No temperature effect was observed on NPP, GPP and R in either season (Table 6; Fig. 5a-c). The high $pCO_2$ condition enhanced NPP in both seasons. The combined effect of season and $pCO_2$ affected GPP, with a positive effect of $pCO_2$ increase in the summer only. Similarly, R significantly increased under high $pCO_2$ in summer conditions. An interactive effect of season and temperature was detected for $G_l$, which increased under high temperature in the summer only (Fig. 5d). Conversely, high $pCO_2$ reduced $G_l$ regardless of the season. In the dark, net dissolution was observed in the winter, but net

precipitation occurred in summer conditions at high temperature (Fig. 5e). $G_d$ was significantly affected by the triple interaction between season, temperature and $pCO_2$. In the winter, high $pCO_2$ increased net dissolutions rates, and high temperature reduced them. In the summer, the interactive effect of temperature and $pCO_2$ increase was more complex, with a decrease in $G_d$ detected under high temperature conditions only.

### 4.    Discussion

Our study demonstrates that the response of maerl bed communities to increased temperature and $pCO_2$ conditions is a complex function of direct effects of climate variables on species physiology and shifts in species interactions. Results show that predicted changes may alter interactions among calcifying and fleshy macroalgae via overgrowth of epiphytic algae and an increase in competition with underlying maerl. Interactions between grazers and macroalgae were also affected because the grazer physiology was adversely affected by acidification and warming with potential consequences on epiphyte biomass

regulation. Our results underscore the importance of examining community-level processes to integrate species interactions in the study of the impact of global change on marine ecosystems.

Assemblage GPP and R were not affected by the high temperature and $pCO_2$ conditions in the winter. Conversely, in the summer, GPP and R increased under high $pCO_2$ conditions. The response of assemblage GPP and R appeared closely related to changes in epiphyte biomass and productivity. For instance, the high biomass of epiphytic algae in the summer led to high





contribution to oxygen fluxes. Under high $pCO_2$ conditions, the higher availability of $CO_2$ as substrate for photosynthesis may stimulate epiphyte productivity and growth (Koch et al., 2013). The two main epiphytic algae that grew during the experiments, *R. ardissonei* and *S. chordalis*, are naturally found in maerl beds in Brittany (Peña et al., 2014). The response of the alga *S. chordalis* to increased temperature and $pCO_2$ differed from that of *R. ardissonei*. This difference suggests that the response is species-specific, even among fleshy algae, as demonstrated by Kram et al. (2016). *R. ardissonei* GPP was not

affected by increased temperature and $pCO_2$, but its R was significantly lower under high $pCO_2$. Within the same genus, Cook et al. (1986) showed that *Rhodymenia palmate* can potentially use $HCO_3^-$ as source of inorganic carbon for photosynthesis. The same process may occur in *R. ardissonei*, suggesting that this alga is not carbon-limited at current oceanic $pCO_2$ levels. In contrast to *R. ardissonei*, increased $pCO_2$ stimulated *S. chordalis* GPP under ambient conditions of temperature. In their study, Short et al. (2014) indicate that the overgrowth of filamentous algae occurs synergistically with

high $pCO_2$ levels and decreased photosynthesis in coralline algae. Here, the stimulation of epiphyte productivity and growth under high $pCO_2$ is likely to increase the competition with underlying maerl, especially through reduction in incident light.

Although assemblages were mainly composed of living and dead maerl, the response of GPP and R of *L. corallioides* to increased temperature and $pCO_2$ differed from that observed in assemblages. For example, the temperature increase of +3°C reduced living *L. corallioides* R in the winter, but increased R in the summer. Under high $pCO_2$ conditions, although $CO_2$

availability for photosynthesis was higher, no difference was observed in *L. corallioides*, probably due to the ability of this species to employ inorganic carbon acquisition mechanisms (Kübler and Dudgeon, 2015). Interestingly, GPP, R and chlorophyll *a* content of dead maerl were of the same magnitude as for living maerl. Although live algae prevent bio-fouling by shedding their surface layers (Keats et al., 1997; Villas Bôas and Figueiredo, 2004), post-mortem colonization by photosynthetic endolithic assemblages may occur within dead crusts (Diaz-Pulido et al., 2012). Moreover, dead thalli may

represent a substrate for the settlement of crustose coralline algae that cover small parts of some thalli. Crustose coralline algae colonization may also contribute to the observed GPP and R values. In dead maerl, only R decreased under high $pCO_2$, while no effect was detected for GPP.

These findings also suggest the importance of dead maerl to assemblage carbonate fluxes during the experiments. For example, endolithic algae appear to play an important role in the dissolution of a crustose coralline alga (CCA) species,



*Porolithon onkodes* (Reyes-Nivia et al., 2014). Through their photosynthesis, endolithic algae may elevate interstitial pH within the *P. onkodes* skeleton (Reyes-Nivia et al., 2013), increasing carbonate cement precipitation (Diaz-Pulido et al., 2014). Within dead *L. corallioides*, the presence of endolithic algae combined with the presence of small patches of CCA on the surface of thalli may explain the calcification rates observed in light and dissolution in dark. Considering the high Mg content in the skeleton of *L. corallioides*, increased $pCO_2$ likely promotes the dissolution of dead thalli. Alternatively, the

increase in dissolution observed in the present study may be associated with a reduction of CCA recruitment over the surface of dead thalli under acidified conditions (Jokiel et al., 2008). These results are consistent with the negative response to increased $pCO_2$ observed here in assemblage $G_l$ and $G_d$ values, which appeared strongly related to the response of living maerl calcification rates. The high sensitivity of coralline algae to ocean acidification has already been attributed to their high Mg-calcite content (Morse et al., 2006; Hofmann and Bischof, 2014). In the present study, the $pCO_2$ increase had

adverse consequences on assemblage $G_d$, both in the winter and summer. In the dark, assemblage R reduced seawater pH by releasing $CO_2$, and hindered the precipitation of $CaCO_3$ (Cornwall et al., 2013). Under high $pCO_2$ conditions, the combined effect of acidification and assemblage R in the dark is likely to increase the sensitivity of living and dead *L. corallioides* to dissolution (Andersson et al., 2009). Moreover, as discussed above, the overgrowth of epiphytic algae under high $pCO_2$ increased assemblage R in the dark. Therefore, the negative effect of ocean acidification on *L. corallioides* $G_d$ would be

exacerbated by the presence of epiphytic algae, which promote a decline in pH in the dark. In light, several studies have suggested that moderate growth of fleshy macroalgal communities may reduce the impact of ocean acidification on coralline calcification by reducing the $CO_2$ concentration of seawater through photosynthesis (Semesi et al., 2009; Short et al., 2014). However, the present findings do not support this idea, because a decline in $G_l$ was observed under high $pCO_2$ despite high epiphyte biomass. Under high $pCO_2$, the overgrowth of epiphytic fleshy algae induced by ocean acidification in the summer

may reduce light, oxygen and nutrient availability for underlying maerl, affecting its primary production and calcification (D'Antonio, 1985; Short et al., 2014). Thus, overgrown maerl would be negatively affected by the direct effect of ocean acidification on calcification rates and indirect effects due to shifts in competition dynamics with fleshy epiphytic algae (Kuffner et al., 2008).



In regard to the present results, the regulation of epiphyte biomass by grazers appears essential to maintain the proper

functioning of maerl bed communities (Guillou et al., 2002). In mollusks and urchins, several studies have demonstrated a

link between feeding rates and other metabolic processes, such as respiration, calcification and excretion (Carr and Bruno,

2013; Navarro et al., 2013; Noisette et al., 2016). In mollusks, a wide range of responses to ocean acidification and warming

have been revealed (Gazeau et al., 2013; Parker et al., 2013). The differences in sensitivity of mollusks to ocean acidification

depend on several parameters, such as the form of $CaCO_3$ they precipitate during calcification (Ries et al., 2009), as well as

their ability to regulate the acid-base balance (Gutowska et al., 2010). Our results corroborate these studies, given that *G.*

*magus* and *J. exasperatus* responded differently to acidification and warming. Increased temperature and $pCO_2$ had no effect

on *G. magus* with regard to the metabolic functions tested. However, despite the apparent resistance of *G. magus* to the

applied changes, other physiological parameters that we did not test here may have been affected, such as feeding rates,

somatic growth, enzyme activity or immune response (Parker et al., 2013). The respiration rates of *J. exasperatus* showed a

decline under high $pCO_2$ in the winter. The lower growth of epiphytes and biofilm in winter may reduce the energy available

to maintain the metabolism under stressful conditions (Thomsen et al., 2013; Pansch et al., 2014). This reduced energy

availability may induce changes in energy partitioning and decrease R under high $pCO_2$. In the summer, the increased R

under high $pCO_2$ can be attributed to higher food supply, which is likely to increase the resistance of *J. exasperatus* to

climate change, as reported for several marine taxa (Ramajo et al., 2016).

Given the relatively high resistance of *G. magus* and *J. exasperatus* to predicted changes, the metabolic response of *P.*

*miliaris* appears to have stronger implications on assemblage functioning. For example, *P. miliaris* is considered as one of

the main macro-epiphytic grazers on maerl beds in the Bay of Brest (Guillou et al., 2002). During the experiments, *P.*

*miliaris* likely played an important role in the regulation of epiphytic biomass. The response of $G_1$ to temperature and $pCO_2$

changes was complex. The interaction between temperature and $pCO_2$ observed in the summer may cause changes in energy

partitioning, thereby inducing a trade-off between metabolic processes at the expense of respiration and excretion (Garilli et

al., 2015). However, the effect of temperature and $pCO_2$ on the calcification of *P. miliaris* must be considered carefully. For

instance, urchins defecated carbonate pellets following consumption of maerl thalli. These feces are likely to dissolve during

incubation, introducing a bias in the measurement of calcification (Gazeau et al., 2015). In the summer, temperature increase



by 3°C reduced *P. miliaris* respiration rates. Moreover, the decrease in excretion under high temperature and pCO$_2$

conditions was modulated by the interaction between these two factors. Temperature is a major factor affecting physiological

processes in ectotherms such as metabolic rates and growth (Kordas et al., 2011). In *P. miliaris*, summer temperatures are

likely to exceed the physiological thresholds of organisms, inducing a metabolic decline when maintained at 20°C. Although

this decline has only been measured for respiration and excretion, the increase in temperature is also likely to affect sea

urchin feeding efficiency (Thomas et al., 2000; Carr and Bruno, 2013). Therefore, the ability of *P. miliaris* to regulate

epiphyte biomass may be significantly altered under predicted acidification and warming conditions.

In addition to the impact of climate change on grazer-fleshy macroalgae interactions, predicted changes may also

considerably alter the interaction between grazers and coralline algae. Asnaghi et al. (2013) demonstrated that the grazing

activity by urchins may exacerbate pCO$_2$ effects on coralline algae. Ocean acidification may alter the structural integrity of

coralline algae, increasing its sensitivity to grazing (Johnson and Carpenter, 2012; Ragazzola et al., 2012). Coralline algae

may thus be more susceptible to grazing by urchins, which also benefit from a higher carbonate uptake from their diet to

modulate their response to ocean acidification (Asnaghi et al., 2013). In *L. corallioides*, the decrease in calcification rates

may alter its structural integrity and increase its susceptibility to grazing, especially by urchins, which are considered as

important bioeroders of coralline algae in marine ecosystems (Ballesteros, 2006; O'Leary and McClanahan, 2010),

particularly in maerl beds (Lawrence, 2013).

In conclusion, the community response to climate change does not appear to be only the result of individual species'

metabolic responses, but also strongly depends on shifts in species interactions. Our results suggest that ocean acidification

and warming will strongly destabilize communities through both direct effects on species physiology and changes in the

interaction strengths between coralline algae, fleshy algae and grazers. Under the predicted business-as-usual conditions,

epiphyte overgrowth may exacerbate the negative impact of climate change on underlying coralline algae. Here, we also

demonstrated that climate change may affect grazer physiology, with major consequences on their ability to regulate

epiphyte biomass. Climate change may also affect other components that we did not assess in the present study, such as algal

palatability and potential changes in grazer trophic behavior (Campbell et al., 2014; Duarte et al., 2015; Poore et al., 2013;

Poore et al., 2016). In line with this study, further work should focus on the impact of climate change on marine communities and species interactions to better understand the consequences on ecosystem functioning.

**Authors' Contributions**

EL SM PR JG JC designed the experiments; EL SM JC collected the data; EL ML analyzed the data; EL SM PR prepared the manuscript with contributions from all co-authors.

**Competing interests**

The authors declare that they have no conflict of interest.

**Acknowledgements**

The authors thank the CRBM (Center of Marine Biologic Resources) at the Station Biologique de Roscoff for its kind permission to use their premises for the duration of the experiments. We are grateful to Olivier Bohner for his laboratory assistance and for the help in system maintenance. We also thank Murielle Jam for freeze-drying samples. We acknowledge the crew of the research vessel *Albert Lucas* for its help with species collection. We also thank the SOMLIT (Service
d'Observation en Milieu LITtoral, INSU-CNRS) program for the temperature data sets provided. We thank Carolyn Engel-Gautier for language editing of the manuscript. This work was supported by the Brittany Regional Council, the French National Research Agency via the "Investment for the Future" program IDEALG (no. ANR-10-BTBR-04), and by the French national EC2CO program ("Écosphère Continentale et Côtière", project MAERLCHANGE).

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

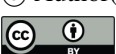




Table 1. Physicochemical parameters (mean ± SE) of seawater in each experimental condition (A-pCO$_2$ = ambient pCO$_2$; H-pCO$_2$ = high pCO$_2$; T = ambient temperature; T+3°C = high temperature) in the winter and the summer. pH$_T$ and temperature were monitored every two days in each aquarium (n = 35). Total alkalinity values (A$_T$) are means (± SE) of 28 samples measured in each aquarium. The CO$_2$ partial pressure (pCO$_2$), dissolved inorganic carbon (DIC), and saturation states of seawater with respect to aragonite ($\Omega_{Ar}$) and calcite ($\Omega_{Ca}$) were calculated from pH$_T$, temperature, salinity, and A$_T$ using CO2SYS.

|  | Experimental condition | pCO$_2$ (µatm) | pH$_T$ | Temperature (°C) | A$_T$ (µmol kg$^{-1}$) | DIC (µmol kg$^{-1}$) | $\Omega_{Ar}$ | $\Omega_{Ca}$ |
|---|---|---|---|---|---|---|---|---|
| **WINTER** | A-pCO$_2$; T | 490 (± 5) | 7.97 (± 0.04) | 10.1 (± 0.3) | 2348 (± 6) | 2189 (± 6) | 1.84 (± 0.02) | 2.89 (± 0.02) |
|  | H-pCO$_2$; T | 1183 (± 10) | 7.63 (± 0.03) | 10.1 (± 0.3) | 2342 (± 7) | 2306 (± 7) | 0.89 (± 0.01) | 1.40 (± 0.01) |
|  | A-pCO$_2$; T+3°C | 513 (± 5) | 7.97 (± 0.03) | 13.7 (± 0.1) | 2341 (± 5) | 2166 (± 5) | 2.01 (± 0.01) | 3.14 (± 0.02) |
|  | H-pCO$_2$; T+3°C | 1087 (± 18) | 7.64 (± 0.03) | 13.6 (± 0.2) | 2329 (± 2) | 2266 (± 4) | 1.09 (± 0.01) | 1.70 (± 0.02) |
| **SUMMER** | A-pCO$_2$; T | 426 (± 4) | 8.03 (± 0.04) | 17.1 (± 0.2) | 2359 (± 3) | 2127 (± 3) | 2.60 (± 0.02) | 4.03 (± 0.03) |
|  | H-pCO$_2$; T | 948 (± 9) | 7.72 (± 0.03) | 17.1 (± 0.2) | 2382 (± 4) | 2279 (± 4) | 1.45 (± 0.01) | 2.24 (± 0.02) |
|  | A-pCO$_2$; T+3°C | 432 (± 4) | 8.01 (± 0.04) | 20.0 (± 0.5) | 2364 (± 3) | 2109 (± 3) | 2.88 (± 0.02) | 4.43 (± 0.03) |
|  | H-pCO$_2$; T+3°C | 879 (± 7) | 7.74 (± 0.02) | 20.2 (± 0.3) | 2369 (± 2) | 2238 (± 2) | 1.71 (± 0.01) | 2.64 (± 0.02) |





Table 2. PERMANOVA results for the effects of season, temperature (T) and $pCO_2$ on respiration, net calcification and excretion rates in the urchin *Psammechinus miliaris* and the two gastropods *Gibbula magus* and *Jujubinus exasperatus* (n = 5). Significant p-values are shown in bold (α = 0.05). Degrees of freedom = 1; F: pseudo F-statistic

| | | Respiration $\mu mol\ O_2\ g\ DW^{-1}\ h^{-1}$ | | Net Calcification $\mu mol\ CaCO_3\ g\ DW^{-1}\ h^{-1}$ | | Excretion $\mu mol\ NH_4^+\ g\ DW^{-1}\ h^{-1}$ | |
|---|---|---|---|---|---|---|---|
| | | F | p-value | F | p-value | F | p-value |
| ***Psammechinus miliaris*** | Season | 257.7 | **<0.001** | 2.8 | 0.11 | 15.2 | **<0.001** |
| | T | 17.1 | **<0.001** | 3.1 | 0.088 | 3.4 | 0.072 |
| | $pCO_2$ | 2.2 | 0.15 | 0.7 | 0.41 | 0.1 | 0.071 |
| | Season x T | 14.7 | **<0.001** | 3.5 | 0.078 | 4.2 | 0.052 |
| | Season x $pCO_2$ | 3.9 | 0.055 | 0.0 | 0.89 | 3.8 | 0.063 |
| | $pCO_2$ x T | 2.3 | 0.14 | 5.6 | **0.025** | 4.6 | **0.037** |
| | Season x T x $pCO_2$ | 0.3 | 0.59 | 5.1 | **0.028** | 0.1 | 0.74 |
| ***Gibbula magus*** | Season | 383.6 | **<0.001** | 26.4 | **<0.001** | 206.7 | **<0.001** |
| | T | 0.1 | 0.72 | 0.1 | 0.72 | 2.7 | 0.11 |
| | $pCO_2$ | 0.2 | 0.64 | 0.6 | 0.43 | 0.6 | 0.43 |
| | Season x T | 0.3 | 0.59 | 0.1 | 0.77 | 1.1 | 0.31 |
| | Season x $pCO_2$ | 1.5 | 0.23 | 0.8 | 0.37 | 2.4 | 0.13 |
| | $pCO_2$ x T | 0.1 | 0.79 | 0.0 | 0.99 | 0.1 | 0.79 |
| | Season x T x $pCO_2$ | 0.2 | 0.66 | 0.3 | 0.62 | 0.7 | 0.41 |
| ***Jujubinus exasperatus*** | Season | 0.9 | 0.35 | | | | |
| | T | 6.5 | **0.017** | | | | |
| | $pCO_2$ | 0.0 | 0.92 | | | | |
| | Season x T | 8.7 | **0.005** | | | | |
| | Season x $pCO_2$ | 14.0 | **<0.001** | | | | |
| | $pCO_2$ x T | 0.8 | 0.38 | | | | |
| | Season x T x $pCO_2$ | 0.4 | 0.54 | | | | |



Table 3. Summary of PERMANOVA for the effects of season, temperature (T) and $pCO_2$ on net and gross primary production, respiration, chlorophyll *a* content and light and dark calcification rates of living and dead *Lithothamnion corallioides* (n = 5). Significant p-values are shown in bold (α = 0.05). Degrees of freedom = 1; F: pseudo F-statistic

### LIVING *L. corallioides*

| | Net production μmol $O_2$ g $DW^{-1}$ $h^{-1}$ | | Gross production μmol $O_2$ g $DW^{-1}$ $h^{-1}$ | | Respiration μmol $O_2$ g $DW^{-1}$ $h^{-1}$ | | Chlorophyll a µg chl a g $DW^{-1}$ | | Light calcification μmol $CaCO_3$ g $DW^{-1}$ $h^{-1}$ | | Dark calcification μmol $CaCO_3$ g $DW^{-1}$ $h^{-1}$ | |
|---|---|---|---|---|---|---|---|---|---|---|---|---|
| | F | p-value | F | p-value | F | p-value | F | p-value | F | p-value | F | p-value |
| Season | 2.3 | 0.14 | 34.3 | **<0.001** | 309.0 | **<0.001** | 0.2 | 0.66 | 85.7 | **<0.001** | 214.8 | **<0.001** |
| T | 4.1 | **0.049** | 2.7 | 0.1 | 0.9 | 0.34 | 0.0 | 0.84 | 6.4 | **0.015** | 4.5 | **0.041** |
| $pCO_2$ | 0.2 | 0.71 | 0.1 | 0.78 | 0.1 | 0.79 | 1.4 | 0.26 | 6.1 | **0.015** | 140.8 | **<0.001** |
| Season x T | 0.1 | 0.78 | 1.5 | 0.22 | 35.7 | **<0.001** | 6.2 | **0.018** | 0.3 | 0.59 | 4.8 | **0.033** |
| Season x $pCO_2$ | 1.1 | 0.31 | 0.8 | 0.39 | 0.1 | 0.76 | 0.0 | 0.88 | 1.7 | 0.22 | 0.2 | 0.63 |
| $pCO_2$ x T | 5.6 | **0.021** | 3.0 | 0.089 | 3.4 | 0.071 | 0.0 | 0.89 | 1.1 | 0.32 | 0.4 | 0.54 |
| Season x T x $pCO_2$ | 0.1 | 0.74 | 0.0 | 0.91 | 0.7 | 0.42 | 0.1 | 0.75 | 0.0 | 0.98 | 0.5 | 0.48 |

### DEAD *L. corallioides*

| | Net production μmol $O_2$ g $DW^{-1}$ $h^{-1}$ | | Gross production μmol $O_2$ g $DW^{-1}$ $h^{-1}$ | | Respiration μmol $O_2$ g $DW^{-1}$ $h^{-1}$ | | Chlorophyll a µg chl a g $DW^{-1}$ | | Light calcification μmol $CaCO_3$ g $DW^{-1}$ $h^{-1}$ | | Dark calcification μmol $CaCO_3$ g $DW^{-1}$ $h^{-1}$ | |
|---|---|---|---|---|---|---|---|---|---|---|---|---|
| | F | p-value | F | p-value | F | p-value | F | p-value | F | p-value | F | p-value |
| Season | 60.8 | **<0.001** | 78.2 | **<0.001** | 115.6 | **<0.001** | 1.8 | 0.18 | 34.4 | **<0.001** | 0.1 | 0.71 |
| T | 3.1 | 0.086 | 3.3 | 0.075 | 2.5 | 0.13 | 0.3 | 0.58 | 4.0 | **0.049** | 0.6 | 0.45 |
| $pCO_2$ | 1.2 | 0.30 | 1.9 | 0.18 | 4.9 | **0.043** | 3.5 | 0.074 | 29.8 | **<0.001** | 44.8 | **<0.001** |
| Season x T | 2.8 | 0.099 | 2.8 | 0.11 | 1.3 | 0.26 | 0.1 | 0.81 | 3.4 | 0.073 | 0.0 | 0.88 |
| Season x $pCO_2$ | 0.6 | 0.46 | 1.2 | 0.29 | 4.0 | 0.053 | 1.4 | 0.26 | 0.2 | 0.69 | 2.0 | 0.16 |
| $pCO_2$ x T | 0.4 | 0.53 | 0.6 | 0.46 | 0.8 | 0.37 | 24.0 | **<0.001** | 2.6 | 0.12 | 0.6 | 0.46 |
| Season x T x $pCO_2$ | 1.1 | 0.31 | 0.7 | 0.40 | 0.0 | 0.94 | 0.2 | 0.68 | 2.9 | 0.11 | 0.6 | 0.45 |





Table 4. Chlorophyll *a* content (mean ± SE) of living and dead *L. corallioides* in the different pCO$_2$ (A-pCO$_2$ = ambient pCO$_2$; H-pCO$_2$ = high pCO$_2$) and temperature (T = ambient temperature; T+3°C = high temperature) treatments, after being maintained three months in winter and summer conditions, n = 5

| | **Chlorophyll *a*** (µg chlorophyll g DW$^{-1}$) | | | | | |
| | A-pCO$_2$/T | H-pCO$_2$/T | A-pCO$_2$/T+3°C | H-pCO$_2$/T+3°C |
| --- | --- | --- | --- | --- |
| **Living *L. corallioides*** | | | | |
| Winter | 59.84 (± 1.97) | 61.66 (± 3.83) | 52.93 (± 3.44) | 56.85 (± 2.52) |
| Summer | 55.03 (± 2.95) | 57.63 (± 3.99) | 60.35 (± 0.70) | 62.19 (± 3.75) |
| **Dead *L. corallioides*** | | | | |
| Winter | 47.09 (± 2.72) | 39.39 (± 5.65) | 39.15 (± 2.20) | 46.36 (± 2.19) |
| Summer | 52.21 (± 1.92) | 36.30 (± 1.83) | 43.63 (± 0.90) | 47.96 (± 2.54) |




Table 5. Summary of the effects of $pCO_2$ and temperature (T) and their combined effect on gross production and respiration of the two epiphytic algae *R. ardissonei* and *S. chordalis* in the summer (n = 5). Statistical analyses were performed using a two-way crossed Scheirer-Ray-Hare test.

Significant p-values are presented in bold ($\alpha$ = 0.05). Degrees of freedom = 1; F: pseudo F-statistic

| | | Net production $\mu mol\ O_2\ g\ DW^{-1}\ h^{-1}$ | | Gross production $\mu mol\ O_2\ g\ DW^{-1}\ h^{-1}$ | | Respiration $\mu mol\ O_2\ g\ DW^{-1}\ h^{-1}$ | |
|---|---|---|---|---|---|---|---|
| | | F | p-value | F | p-value | F | p-value |
| *Rhodymenia ardissonei* | T | 0.8 | 0.37 | 0.2 | 0.68 | 1.3 | 0.25 |
| | $pCO_2$ | 0.0 | 0.96 | 0.8 | 0.38 | 8.6 | **0.003** |
| | $pCO_2$ x T | 1.0 | 0.31 | 1.0 | 0.31 | 0.7 | 0.42 |
| *Solieria chordalis* | T | 0.1 | 0.76 | 0.1 | 0.80 | 5.5 | **0.019** |
| | $pCO_2$ | 3.0 | 0.08 | 8.3 | **0.011** | 3.9 | **0.049** |
| | $pCO_2$ x T | 5.8 | **0.016** | 7.5 | **0.014** | 0.0 | 0.48 |

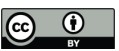

Table 6. Summary of PERMANOVA for the effects of season, temperature (T) and $pCO_2$ on net and gross primary production, respiration and light and dark calcification rates, measured on assemblages (n = 5). Significant p-values are presented in bold ($\alpha$ = 0.05). Degrees of freedom = 1;

F: pseudo F-statistic

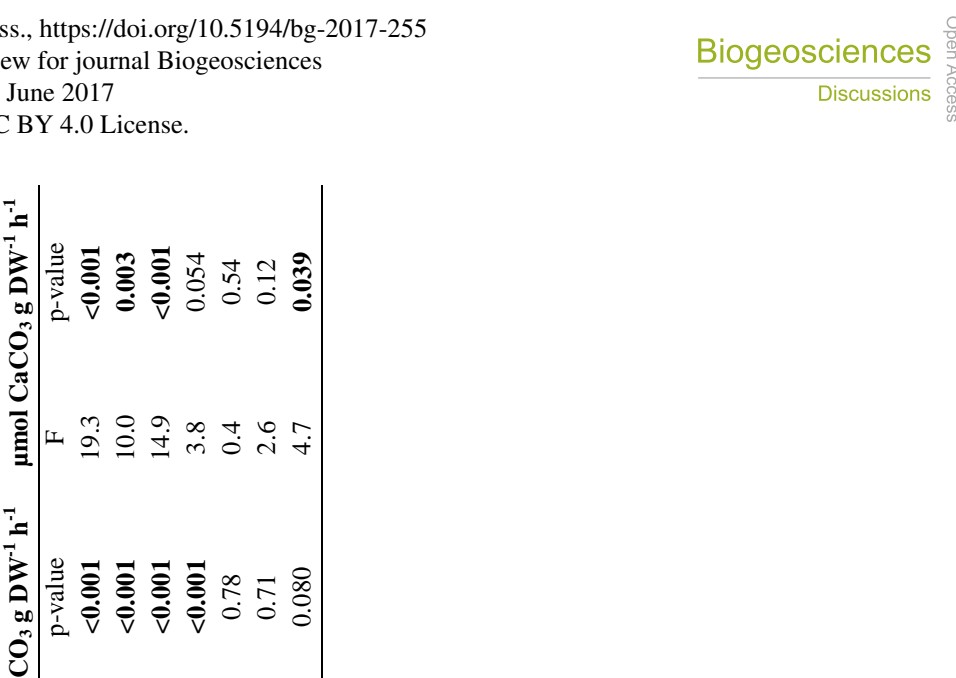

| | Net production $\mu$mol O$_2$ g DW$^{-1}$ h$^{-1}$ | | Gross production $\mu$mol O$_2$ g DW$^{-1}$ h$^{-1}$ | | Respiration $\mu$mol O$_2$ g DW$^{-1}$ h$^{-1}$ | | Light calcification $\mu$mol CaCO$_3$ g DW$^{-1}$ h$^{-1}$ | | Dark calcification $\mu$mol CaCO$_3$ g DW$^{-1}$ h$^{-1}$ | |
|---|---|---|---|---|---|---|---|---|---|---|
| | F | p-value | F | p-value | F | p-value | F | p-value | F | p-value |
| Season | 30.2 | **<0.001** | 50.6 | **<0.001** | 219.7 | **<0.001** | 290.5 | **<0.001** | 19.3 | **<0.001** |
| T | 1.2 | 0.28 | 2.2 | 0.14 | 1.7 | 0.19 | 38.1 | **<0.001** | 10.0 | **0.003** |
| pCO$_2$ | 15.7 | **<0.001** | 9.6 | **0.004** | 2.1 | 0.16 | 46.0 | **<0.001** | 14.9 | **<0.001** |
| Season x T | 1.7 | 0.21 | 0.2 | 0.66 | 0.1 | 0.73 | 15.7 | **<0.001** | 3.8 | 0.054 |
| Season x pCO$_2$ | 0.6 | 0.43 | 7.1 | **0.013** | 10.9 | **0.001** | 0.1 | 0.78 | 0.4 | 0.54 |
| pCO$_2$ x T | 2.7 | 0.11 | 0.4 | 0.53 | 0.1 | 0.72 | 0.2 | 0.71 | 2.6 | 0.12 |
| Season x T x pCO$_2$ | 0.4 | 0.55 | 1.2 | 0.29 | 1.3 | 0.27 | 3.3 | 0.080 | 4.7 | **0.039** |

Assemblages



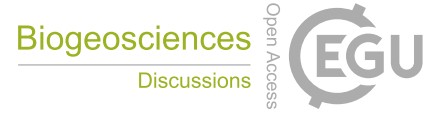

Fig. 1. Respiration, net calcification and excretion rates (mean ± SE) of the grazers *P. miliaris* (a to c), *G. magus* (d to f) and respiration of *J. exasperatus* (g) in the different $pCO_2$ (A-$pCO_2$ = Ambient $pCO_2$; H-$pCO_2$ = High-$pCO_2$) and temperature (T = Ambient temperature; T+3°C = High temperature) conditions. The species were maintained in assemblages for three

months in winter (dark gray) and summer conditions (light gray). n = 5




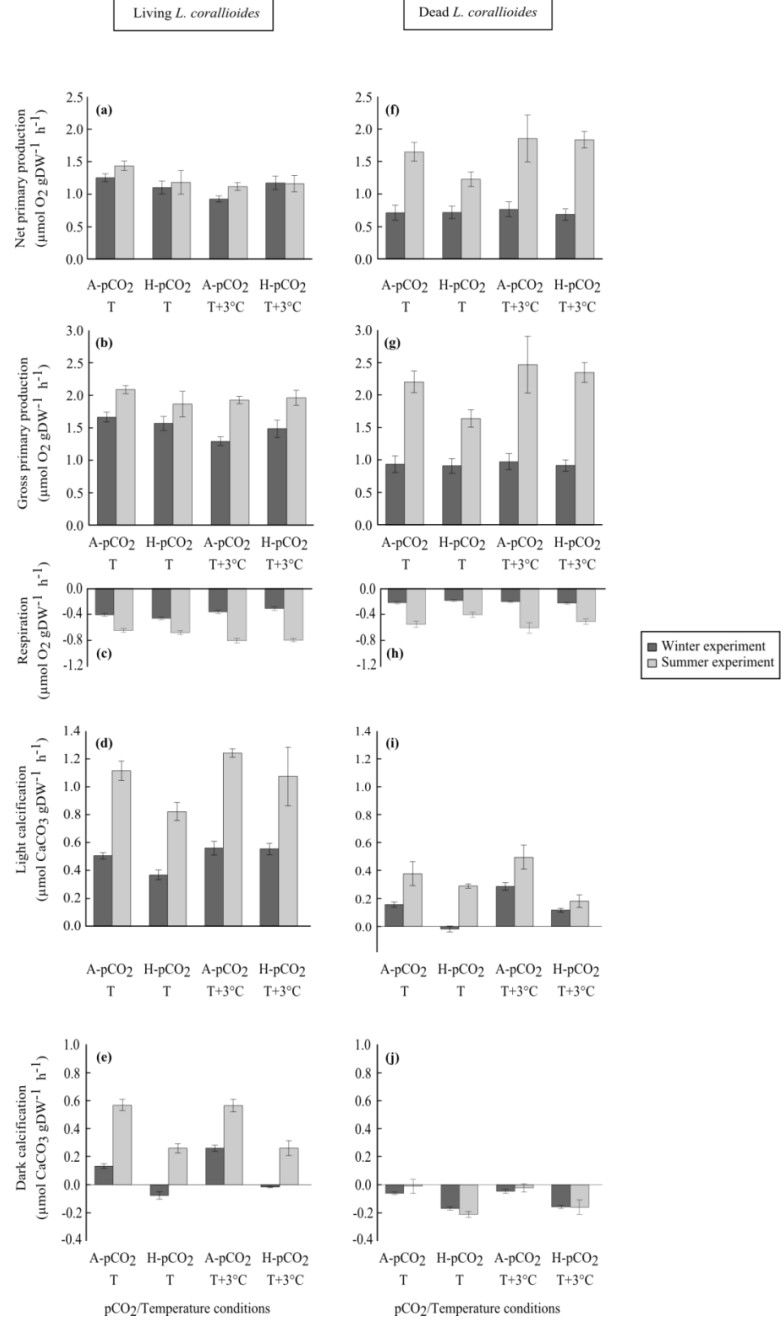

Fig. 2. Net and gross primary production, respiration, light and dark calcification rates (mean ± SE) of living (a to e) and dead thalli (f to j) of *L. corallioides* in the different $pCO_2$ (A-$pCO_2$ = Ambient $pCO_2$; H-$pCO_2$ = High-$pCO_2$) and temperature (T = Ambient temperature; T+3°C = High temperature) treatments, after three months in winter (dark gray) and summer conditions (light gray). n = 5





Fig. 3. Summer net and gross primary production and respiration rates (mean ± SE) of the two main epiphytic fleshy algae *Rhodymenia ardissonei* (a to c) and *Solieria chordalis* (d to f), in the different $pCO_2$ (A-$pCO_2$ = Ambient $pCO_2$; H-$pCO_2$ = High-$pCO_2$) and temperature (T = Ambient temperature; T+3°C = High temperature) treatments. n = 5





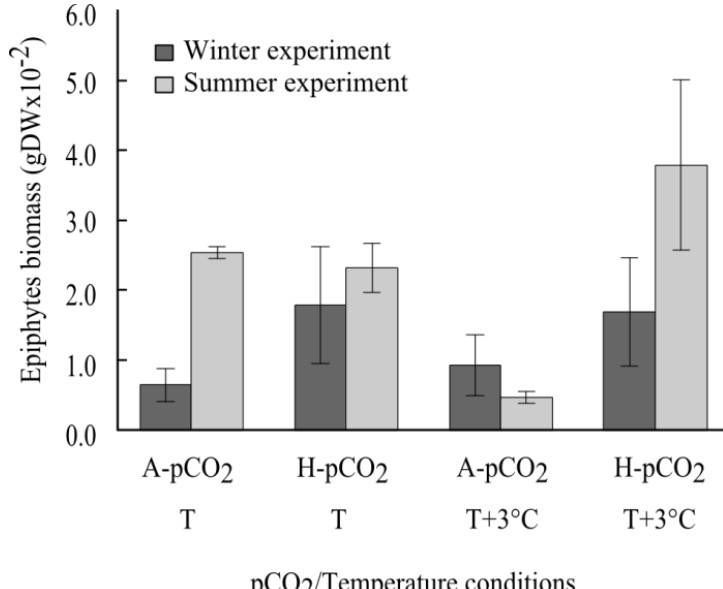

Fig. 4. Biomass of epiphytic fleshy algae obtained in the different $pCO_2$ (A-$pCO_2$ = Ambient $pCO_2$; H-$pCO_2$ = High-$pCO_2$) and temperature (T = Ambient temperature; T+3°C = High temperature) treatments, after the three-month experiments in

winter (dark gray) and summer (light gray) experiments. n = 5




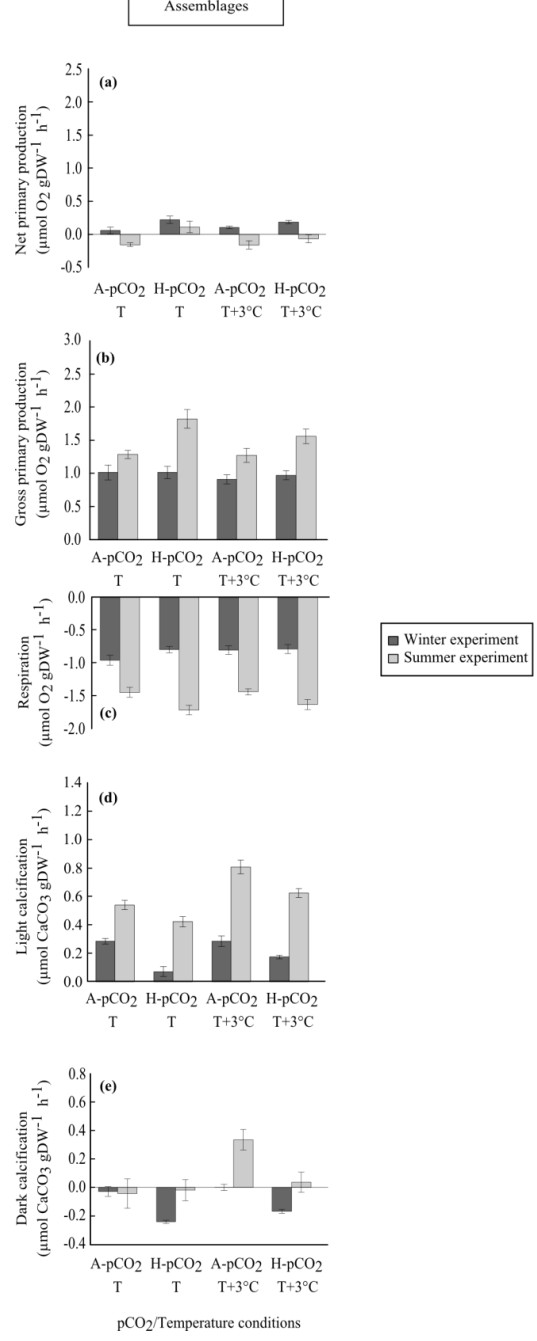

Fig. 5. Net and gross primary production (a and b, respectively), respiration (c) and light and dark calcification rates (d and e, respectively) rates (mean ± SE) of assemblages in the different $pCO_2$ (A-$pCO_2$ = Ambient $pCO_2$; H-$pCO_2$ = High-$pCO_2$) and temperature (T = Ambient temperature; T+3°C = High temperature) treatments. The assemblages were maintained during three months in winter (dark gray) and summer conditions (light gray). n = 5