# Peer review of "Species interactions can shift the response of a maerl bed community to ocean acidification and warming"

_Biogeosciences, 2017_

## Referee Comment (RC1) · W. R. Hunter (Referee) · 18 Jul 2017

Review of Legrand et al. Species interactions can shift the response of a maerl bed community to ocean acidification and warming

**General Comments**

The paper by Legrand et al. describes an elegant mesocosm experiment testing the effects of ocean warming and acidification upon the community-scale responses of maerl bed assemblages. This work is timely, and provides an interesting insight into how the communities associated with coralline algae will respond to the impacts of anthropogenic climate change. Given the importance of coralline algae as a habitat architect, and the role of these communities in carbon fixation, I believe this work will make an important contribution to our understanding of coastal sea biogeochemistry. The statistical analysis, however, leaves a lot to be desired and as such, I cannot confidently review the authors' interpretation of their results or discussion. I am baffled as to why the authors have chosen to use permutational multivariate analysis of variance (perMANOVA) of similarity matrices (Euclidean distance) as a statistical test to test for differences in univariate response variable (e.g. respiration). Firstly, the authors make the erroneous assumption that perMANOVA does not make any assumptions about normality and homoscedacity of the data. However. Anderson (2001) point out in their papers describing these methods that the method makes the assumption of multivariate normality as measured by a homogenous dispersion of the similarity matrix data. Secondly, I simply cannot understand why perMANOVA was selected as a statistical test. There are more appropriate univariate tests such as analysis of variance (ANOVA) [with appropriate transformations applied], or if appropriate the use of Generalised Linear Models or Generalised Least Squares techniques which would allow the author to account for non-Gaussian data distributions (GLM) or heterogeneous variances between the treatments (GLS) (see Zuur et al., 2009). This represents a major issue with the handling of the experimental data, and so I cannot recommend the paper be accepted for publication in its current form. I strongly encourage the authors to revise the paper and resubmit. I think this has the potential to be an excellent paper and I will happily review a suitably revised manuscript.

**Specific Comments**

*Abstract*

Pg. 1 L 11: "However, little information is available on the response of marine communities..." I do not believe this is true. There has been considerable work of community scale responses to OA – see Ulf Riebesell's work on planktonic communities and bentho-pelagic coupling as an example.

*Introduction*

Pg. 2 L 34-35: Please specify examples of how species interactions are modified by climate change.

Pg. 2 L 37: There are actually quite a number of studies examining the effects of climate change on marine communities. I recommend the authors carry out a thorough literature search.

Pg. 3 L 61-63: "Because the responses of species..." This sentence seems rather poorly structured consider revising to clarify.

*Materials and Methods*

Pg. 4 L 90 – 97: This should be a single paragraph.

Pg. 5 L 100-109: This information would be better displayed as a table.

Pg. 9 L 190 – 201: Please revise around appropriate statistical tests.

*Discussion*

Pg. 11 L 251-253: "Results show… underlying maerl." This sentence is not clear, please specify the community responses to climate change more clearly.

Pg. 16 L 358-359: The final line of the paper is vague, what specific pieces of further work would be useful?

*Figures*

In the figures it would helpful to see which treatment effects are statistically significant, can you please find a way to highlight these effects in the graphs.

**References**

Anderson, M.J. 2001. A new method for non-parametric multivariate analysis of variance. Austral Ecology, 26: 32–46.

Zuur, A. et al., 2009. Mixed Effects Models and Extensions in Ecology with R. Springer-Verlag. 574pp. DOI: 10.1007/978-0-387-87458-6.

---

## Referee Comment (RC2) · L. Hofmann (Referee) · 28 Jul 2017

General Comments

The manuscript "Species interactions can shift the response of a maerl bed community to ocean acidification and warming" describes a novel experiment with an interesting approach on community interactions under predicted global climate change that are generally lacking in the literature. Considering ocean acidification and warming are occurring simultaneously and interdependently, experiments that investigate the effects of both factors on marine organisms are important for understanding future changes in physiology and ecology. The authors were able to do this in their study, and not

only did they investigate effects of ocean warming and acidification on the physiology of single organisms, but also of communities. Through their experimental design, they are able to describe changes in species interactions under future climate change conditions, which is currently rare in the literature. The experimental design is good and the manuscript is well written thorough. My main criticism is that the results could be described more clearly and thoroughly. The interactions between the independent variables should be described more clearly. Interaction plots could help with the interpretation of the statistical analysis of the effect of season, temperature and CO2 on the independent variables. The authors tested the effect of season on the dependent variables, but they often fail to describe this effect in the results section, and focus only on the CO2 and temperature effect. They also fail to mention in some cases that temperature ameliorates the negative effect of pCO2 on some variables, which is important considering both warming and acidification are occurring interdependently. I have made specific comments below.

Specific Comments

Materials and Methods

Line 141-142 "Before incubation, epiphytic algae that spontaneously grew on L. corallioides during the experiments were carefully removed and incubated separately." I assume this was done after the assemblage measurements were made? The authors could clarify this here.

Line 156 What buffer solutions were used to calibrate the reactive spots?

Results

In the results headings, the authors mention acidification and warming, but ignore the factor season

Lines 205-208 I think the results can be described more thoroughly here. There actually was not a negative effect of CO2 and temperature on Gl compared to the control.

Temperature increased calcification rates in the summer. CO2 alone did not seem to have an affect in either season. The combination of high CO2 and temperature in summer negated the positive effect of temperature. The authors should mention there was a main effect of season on P. miliaris E. Excretion was highest under control conditions in the summer. High temperature, CO2 and the combination of both decreased excretion rates in the summer.

Lines 209-210 It is confusing to say R was positively or negatively affected - please rather describe if it increased or decreased. Also, although there was no temperature or CO2 affect on R, Gl, or E, there was a strong effect of season.

Line 213 Please add the effect of season, e.g. "Living maerl GPP did not differ among temperature and pCO2 conditions, but there was a strong effect of season, with higher rates in the summer than in the winter."

Line 215 Add the effect of season on chlorophyll a

Line 216 "Temperature had a positive effect on the Gl of living maerl. Conversely, Gl was significantly reduced under high pCO2..." The authors fail to mention that in the combined treatment, temperature alleviated the negative effect of pCO2. This is very important to the story.

Line 220 "Net dissolution, because Gd was negative, was recorded in the winter under high pCO2 conditions" But dissolution was less in the combined temperature + CO2 treatment in the winter, so temperature alleviated some of the negative effect of CO2 in the winter, although net dissolution still occurred.

Line 222 Again, mention the main affect of season

Line 223 I did not see an interaction between season and pCO2 for GPP in Table 3

Line 225 Mention the effect of season on dead maerl

Line 233 "R was enhanced by the high temperature and pCO2 conditions..." alone, and

their combination resulted in the greatest R rates.

Line 238 Add that temperature alone decreased epiphyte biomass in the summer.

Line 240 "No temperature effect was observed...." But all response variables were higher in the summer than in the winter.

Line 248 "In the summer, the interactive effect of temperature and pCO2 increase was more complex, with a (change to) increase in Gd detected under high temperature conditions only."

Discussion

The authors state that "ocean acidification and warming will strongly destabilize communities through both direct effects on species physiology and changes in the interaction strengths between coralline algae, fleshy algae, and grazers." Based on the assemblage data, I do not think that the effect is so negative, at least in the summer. There is a strong difference in the effect of the combination of CO2 and temperature in winter and summer. In summer, assemblages exposed to high temperature and pCO2 combined actually had similar to or even slightly higher light calcification rates than the ambient treatment. In winter, there was a decrease in light calcification compared to the ambient treatment, but the positive effect of temperature and the negative effect of pCO2 were weakened when the two were combined. I think it is important for the authors to point out that the combination of pCO2 and temperature often subdued the effects of each single factor, because it illustrates the point that experiments investigating only the effect of pCO2 or temperature may present more dramatic responses than when the two are combined, which represents a more realistic scenario.

Technical Comments

Line 129: insert "the" before CO2SYS

It would be helpful to be able to identify statistically significant differences in the figures

---

## Referee Comment (RC3) · G. Diaz-Pulido (Referee) · 14 Aug 2017

Review of the manuscript by LEGRAND et al. entitled "Species interactions can shift the response of a maerl bed community to ocean acidification and warming", submitted to Biogeosciences.

The study by Legrand et al. assessed the metabolic responses of a range of species associated to maerl beds (incl coralline algae, grazers and epiphytic fleshy algae), as well as the metabolic responses of the maerl assemblage to changes in seawater carbonate chemistry and temperature across two climatic seasons. The authors found complex interactions among experimental factors and seasons on the species and

community metabolism. The coralline algae exhibited responses which were expected under CO2 perturbation experiments, but importantly, the study documented significant changes to grazers' metabolism and enhanced epiphytic algal biomass under CO2 enrichment. Although ecological interactions were not directly assessed, changes in the metabolic responses of the experimental species are assumed to influence species interactions. Based on these results the authors were able to propose that ocean acidification and warming will have considerable impacts on the functioning of maerl beds.

I read this manuscript with great interest and believe the authors have done a comprehensive and thorough study. Most studies in the field of impacts of climate change on marine systems focus on responses of one or two species, generally within the same taxonomic group, and it is refreshing to see that this study took a step forward and assessed the impacts at the community level during two climatic seasons. Individual responses focussed on a range of response variables, incl chlorophyll (for the algae), net production, respiration, net calcification (light and dark), and excretion for the grazers. In combination with the assemblage's responses, this allowed the authors to discuss some potential ecological implication such as shifts in species composition, competition, carbon storage, etc. The Methods are generally well described and provide enough detail so that other researchers can repeat the experiments. Methods are appropriate for ocean acidification research.

Main comments: I have two main comments to the paper. First, seasonal effects on both the individual and assemblage responses were not fully explored or discussed in the m/s. One of the strengths of this m/s is that it was conducted in two different climatic seasons, but how the strength of the responses varied between seasons was not clear. I would suggest that the authors include a section where this comment can be fully addressed.

The statistical analyses seem to be well executed, however, I would argue that because there were significant interactions between treatments (OA, temp, and season), there is a need to conduct further statistical analyses within treatment combinations, as in

several instances, the main factor was significant, but in fact it was only significant for one or the other season, or under a particular treatment combination. For example, in line 213, "R was significantly reduced by the high temperature condition in the winter, whereas an increase in R was observed in the summer." This statement is fine, but is not actually supported by a statistical analysis as Table 3 only provides p values for the main effects. This issue is also evident 216-219. Underwood (1997; Experiments in Ecology: Their Logical Design and Interpretation Using Analysis of Variance, Cambridge University Press) provides information on this topic. These new analyses could be included as supplementary material.

There are some statements that are not supported by the experiments. Although the authors demonstrated changes in algal and grazer metabolisms, species interactions among those organisms were not examined experimentally. E.g. Line 251. "Our study demonstrates that the response of maerl bed communities to increased temperature and pCO2 conditions is a complex function of direct effects of climate variables on species physiology and shifts in species interactions". Reword this statement.

Minor comments: • Unclear why chla was measured on dead Lithothamnion. Provide a brief justification in section 3.3. • Line 90: In general avoid single-sentence paragraphs. • Line 237: ".. having positive effect". Was this effect significant? • L260-280: This is a very long paragraph, try breaking it into two. • 285-305: This is also a very long paragraph. • 291: Ordonez et al. (Ordonez Alvarez et al. 2014 Effects of ocean acidification on population dynamics and community structure of crustose coralline algae. Biological Bulletin 226, 255-268.) also found a failure in recruitment of tropical CCA and importantly documented shifts in species composition.

• Line 303: "However, the present findings do not support this idea, because a decline in Gl was observed under high pCO2 despite high". Short et al (2014) paper dealt with minute algal turfs which may have altered the thickness of the diffusive boundary layer on the coralline algae. The macroalgae investigated in the present study were much bigger and may interact in many different ways. It is perhaps very difficult to

generalise the impacts of epiphytic algae on coralline algae given the diversity of algae in marine systems. Perhaps a line or two addressing this would be useful.

• Pages 14-15: Grazing responses may also be altered by changes in seaweed allelopathic compounds, brought about by changes in composition, quantity, or in the magnitude/potency of the allelopathic interactions. A recent study showed that the potency of allelopathic interactions towards a tropical coral was intensified under ocean acidification conditions (Del Monaco et al. 2017 Effects of ocean acidification on the potency of macroalgal allelopathy to a common coral. Scientific Reports 7, 41053). May be worth adding this potential mechanism as drivers of changes in species interactions in response to acidification and warming.

---

## Author Comment (AC1) · 20 Sep 2017

**Review of Legrand et al. Species interactions can shift the response of a maerl bed community to ocean acidification and warming**

We thank the referee for the constructive comments. We considered all the suggestions and improved the manuscript accordingly. Answers are in red color.

**General Comments**

The paper by Legrand et al. describes an elegant mesocosm experiment testing the effects of ocean warming and acidification upon the community-scale responses of maerl bed assemblages. This work is timely, and provides an interesting insight into how the communities associated with coralline algae will respond to the impacts of anthropogenic climate change. Given the importance of coralline algae as a habitat architect, and the role of these communities in carbon fixation, I believe this work will make an important contribution to our understanding of coastal sea biogeochemistry. The statistical analysis, however, leaves a lot to be desired and as such, I cannot confidently review the authors' interpretation of their results or discussion. I am baffled as to why the authors have chosen to use permutational multivariate analysis of variance (perMANOVA) of similarity matrices (Euclidean distance) as a statistical test to test for differences in univariate response variable (e.g. respiration). Firstly, the authors make the erroneous assumption that perMANOVA does not make any assumptions about normality and homoscedacity of the data. However. Anderson (2001) point out in their papers describing these methods that the method makes the assumption of multivariate normality as measured by a homogenous dispersion of the similarity matrix data. Secondly, I simply cannot understand why perMANOVA was selected as a statistical test. There are more appropriate univariate tests such as analysis of variance (ANOVA) [with appropriate transformations applied], or if appropriate the use of Generalised Linear Models or Generalised Least Squares techniques which would allow the author to account for non-Gaussian data distributions (GLM) or heterogeneous variances between the treatments (GLS) (see Zuur et al., 2009). This represents a major issue with the handling of the experimental data, and so I cannot recommend the paper be accepted for publication in its current form. I strongly encourage the authors to revise the paper and resubmit. I think this has the potential to be an excellent paper and I will happily review a suitably revised manuscript.

Answer: As suggested, the statistical design has been modified in the m/s and is described in the section "2.5. Data analysis" (P. 9 Lines 191-197):

"Comparisons in species and assemblage physiological rates between the winter and summer seasons was performed using t-tests, after checking the normality and homogeneity of variances. The influence of temperature and $pCO_2$ was tested on metabolic rates of grazers (*P. miliaris*, *G. magus* and *J. exasperatus*), living and dead maerl, epiphytic biomass and assemblages. Normality of the data and variance homogeneity were checked for all variables. When assumptions were respected, two-way ANOVA were performed, using temperature and $pCO_2$ as fixed orthogonal factors. When assumptions were not respected, two-way non-parametric Scheirer-Ray-Hare tests were run. Statistical analyses were conducted separately for winter and summer experiments in order to keep a balanced design."

**Specific Comments**

*Abstract*

Pg. 1 L 11: "However, little information is available on the response of marine communities…" I do not believe this is true. There has been considerable work of community scale responses to OA – see Ulf Riebesell's work on planktonic communities and bentho-pelagic coupling as an example.

A: We have specified "benthic communities" in the abstract. (L. 11)

*Introduction*

Pg. 2 L 34-35: Please specify examples of how species interactions are modified by climate change.

A: "Species interactions are a key element in ecosystem functioning and are likely to attenuate or amplify the direct effects of climate change on individual species (O'Connor et al., 2011; Hansson et al., 2012; Kroeker et al., 2012)." (L. 35-36)

Pg. 2 L 37: There are actually quite a number of studies examining the effects of climate change on marine communities. I recommend the authors carry out a thorough literature search.

A: We have reworded the sentence to reflect the growing interest of researches on benthic communities: (L. 36-39) "Most research on benthic ecosystems has focused on the impact of ocean acidification and warming on the response of single species (Yang et al., 2016) and

despite a growing interest, studies examining the effects of climate change at the community scale are scarce in the literature (Hale et al., 2011; Alsterberg et al., 2013)."

Pg. 3 L 61-63: "Because the responses of species…" This sentence seems rather poorly structured consider revising to clarify.

A: The sentence has been clarified: (L. 62-64) "Because the response of species and communities to climate change is likely to vary depending on seasonal changes in environmental factors, such as light intensity, photoperiod and temperature (Godbold and Solan, 2013; Martin et al., 2013; Baggini et al., 2014), it was tested in both winter and summer conditions."

*Materials and Methods*

Pg. 4 L 90 – 97: This should be a single paragraph.

A: Done

Pg. 5 L 100-109: This information would be better displayed as a table.

A: A new table (Table 1) shows the different $pCO_2$ and temperature conditions used for winter and summer experiments.

Table 1. Summary of the four experimental treatments. Two $pCO_2$ (ambient and high $pCO_2$) and temperature (ambient and high temperature) conditions were tested. High $pCO_2$ (H-$pCO_2$) corresponded to a pH decrease of -0.33 units compared to ambient conditions (A-$pCO_2$). High temperature (T + 3°C) corresponded to a temperature increase of 3°C compared to ambient conditions (T).

|  | $pCO_2$ | Temperature | |
| --- | --- | --- | --- |
| 1 (Control) | Ambient (A-$pCO_2$) | Ambient (T) | A-$pCO_2$; T |
| 2 | High (H-$pCO_2$) | Ambient (T) | H-$pCO_2$; T |
| 3 | Ambient (A-$pCO_2$) | High (T+3°C) | A-$pCO_2$; T + 3°C |
| 4 | High (H-$pCO_2$) | High (T+3°C) | H-$pCO_2$; T + 3°C |

Pg. 9 L 190 – 201: Please revise around appropriate statistical tests.

A: The statistical design has been changed as discussed above.

*Discussion*

Pg. 11 L 251-253: "Results show… underlying maerl." This sentence is not clear, please specify the community responses to climate change more clearly.

A: (L. 254-256) "Results show that predicted changes may alter interactions among calcifying and fleshy macroalgae via overgrowth of epiphytic algae and an increase in competition for light and nutrients with underlying maerl."

Pg. 16 L 358-359: The final line of the paper is vague, what specific pieces of further work would be useful?

A: The sentence has been reworded: (L. 378-381) "In order to better understand the consequences of climate change on ecosystem functioning, further work should focus on the response of marine communities and consider more specifically shifts in species interactions, including changes in trophic interactions between algae and grazers."

*Figures*

In the figures it would helpful to see which treatment effects are statistically significant, can you please find a way to highlight these effects in the graphs.

A: Following the suggestion of Referees #1 and #2, statistically significant results have been added on graphs: (L. 197-198) "When 2-way AVNOVAs showed significant results, post hoc tests (Tukey honest significant difference, HSD) were performed to compare the four treatments." Results have been added on corresponding graphs. The direction of changes have also been added in tables (Tables 4, 5, 7 and 8) and interaction plots (in supplementary material) when a significant interaction between $pCO_2$ and temperature was detected.

---

## Author Comment (AC2) · 20 Sep 2017

**General Comments**

We thank the referee for the thoughtful and constructive comments that helped to improve the manuscript. We considered all the suggestions and improved the manuscript accordingly. Answers to referee's comments are in red color.

The manuscript "Species interactions can shift the response of a maerl bed community to ocean acidification and warming" describes a novel experiment with an interesting approach on community interactions under predicted global climate change that are generally lacking in the literature. Considering ocean acidification and warming are occurring simultaneously and interdependently, experiments that investigate the effects of both factors on marine organisms are important for understanding future changes in physiology and ecology. The authors were able to do this in their study, and not only did they investigate effects of ocean warming and acidification on the physiology of single organisms, but also of communities. Through their experimental design, they are able to describe changes in species interactions under future climate change conditions, which is currently rare in the literature. The experimental design is good and the manuscript is well written thorough.

My main criticism is that the results could be described more clearly and thoroughly. The interactions between the independent variables should be described more clearly. Interaction plots could help with the interpretation of the statistical analysis of the effect of season, temperature and CO2 on the independent variables. The authors tested the effect of season on the dependent variables, but they often fail to describe this effect in the results section, and focus only on the CO2 and temperature effect. They also fail to mention in some cases that temperature ameliorates the negative effect of pCO2 on some variables, which is important considering both warming and acidification are occurring interdependently. I have made specific comments below.

A: The statistics have been changed according to the comment of Reviewer #1. The seasonal effect has been analyzed separately using t-tests. The effect of increased $pCO_2$ and temperature on the metabolism of species and assemblage was examined in the winter and the summer using 2-way ANOVA. When an interactive effect of $pCO_2$ and temperature was evidenced, interaction plots were performed and provided in supplementary material to this

paper. As suggested by the reviewer, the seasonal effect on metabolic parameters has now been discussed in the discussion section of the revised manuscript (L. 263-267). We also considered more closely the importance of season in the response of organisms and assemblages to acidification and warming.

[Figure]

Supplementary material. Interaction plots for the effects of temperature and $pCO_2$ on dead maerl chlorophyll *a* content in (a) the summer and (f) winter seasons, (b) *P. miliaris* net calcification in the summer, *S. chordalis* (c) net and (f) gross primary production in the

summer, and (e) epiphytes biomass in the summer. Plots were done only when an interactive effect of temperature and $pCO_2$ was detected using 2-way ANOVA (p-value in bold).

**Specific Comments**

*Materials and Methods*

Line 141-142 "Before incubation, epiphytic algae that spontaneously grew on L. corallioides during the experiments were carefully removed and incubated separately." I assume this was done after the assemblage measurements were made? The authors could clarify this here.

A: Assemblage incubation was performed first. After this, epiphytic algae were removed from maerl. The sentence has been modified: (L. 137-138) "After assemblage incubations, epiphytic algae that spontaneously grew on *L. corallioides* during the experiments were carefully removed and incubated separately."

Line 156 What buffer solutions were used to calibrate the reactive spots?

A: (L. 152-154) "The 0% buffer solution was prepared by dissolving 1 g of sodium sulfite ($Na_2SO_3$) in 100 mL of seawater. The 100% buffer solution was prepared by bubbling air into 100 mL of seawater using an air-pump for 20 min to obtain air-saturated seawater." This information has been added in the revised m/s.

*Results*

In the results headings, the authors mention acidification and warming, but ignore the factor season

A: The factor season has now been taken into account for grazers (L. 203-205), living maerl (L. 214), dead maerl (L. 225-226), epiphyte biomass (L. 240-241) and assemblages (L. 245-246).

Lines 205-208 I think the results can be described more thoroughly here. There actually was not a negative effect of CO2 and temperature on Gl compared to the control. Temperature increased calcification rates in the summer. CO2 alone did not seem to have an effect in either season. The combination of high CO2 and temperature in summer negated the positive effect of temperature.

A: The main effect of season on *P. miliaris* has been mentioned (L. 203-204). The sentence on the effect of temperature and $pCO_2$ on *P. miliaris* $G_l$ has been reworded: (L. 206-207) "*P. miliaris* $G_l$ was significantly affected by the interaction between temperature and $pCO_2$ in the summer (Fig. 1b, supplementary material b), which negated the positive effect of increased temperature and $pCO_2$ alone.". We also used interaction plots (in supplementary material) to illustrate $pCO_2$ and temperature combined effect.

The authors should mention there was a main effect of season on P. miliaris E. Excretion was highest under control conditions in the summer. High temperature, CO2 and the combination of both decreased excretion rates in the summer.

A: We have added the effect of season on *P. miliaris* E. We have reworded this section: (L. 208-209) "*P. miliaris* E was higher under control conditions in the summer and increased temperature significantly reduced *P. miliaris* E (Table 4; Fig. 1c)."

Lines 209-210 It is confusing to say R was positively or negatively affected - please rather describe if it increased or decreased. Also, although there was no temperature or CO2 affect on R, Gl, or E, there was a strong effect of season.

A: We agree with the Reviewer and changed the sentence accordingly: (L. 210-211) "In *J. exasperatus*, R increased under elevated temperature but in winter conditions only (Table 4; Fig. 1g)." The effect of season has now been added.

Line 213 Please add the effect of season, e.g. "Living maerl GPP did not differ among temperature and pCO2 conditions, but there was a strong effect of season, with higher rates in the summer than in the winter."

A: The effect of season has now been added: (L. 214) "The metabolism of living *L. corallioides* was higher in the summer than in the winter, except for NPP (Table 3)."

Line 215 Add the effect of season on chlorophyll a

A: We have added: (L. 217-218) "No effect of season was observed on chlorophyll *a* content (Tables 3; 6)."

Line 216 "Temperature had a positive effect on the Gl of living maerl. Conversely, Gl was significantly reduced under high pCO2..." The authors fail to mention that in the combined

treatment, temperature alleviated the negative effect of pCO2. This is very important to the story.

A: The sentence has been revised due to the change in statistical design. (L. 218-219) "The $G_l$ of living maerl was not significantly influenced by increased temperature and $pCO_2$, regardless of the season"

Line 220 "Net dissolution, because Gd was negative, was recorded in the winter under high pCO2 conditions" But dissolution was less in the combined temperature + CO2 treatment in the winter, so temperature alleviated some of the negative effect of CO2 in the winter, although net dissolution still occurred.

A: A sentence has been added: (L. 222-223) "This negative effect of increased $pCO_2$ was alleviated under elevated temperature."

Line 222 Again, mention the main effect of season

A: The effect of season has now been added (L. 225-226).

Line 223 I did not see an interaction between season and pCO2 for GPP in Table 3 Line 225 Mention the effect of season on dead maerl

A: We apologize for this mistake, this has been withdrawn. The sentence has been modified to consider the effect of season (L. 225-226).

Line 233 "R was enhanced by the high temperature and pCO2 conditions..." alone, and their combination resulted in the greatest R rates.

A: We have added this information: (L. 238-239) "R was enhanced by the high temperature and $pCO_2$ conditions and their combination resulted in a greater R"

Line 238 Add that temperature alone decreased epiphyte biomass in the summer.

A: We have added "Epiphyte biomass was not affected by increased temperature or $pCO_2$ in the winter (2-way ANOVA, p=0.95 and 0.67 respectively), while an interactive effect of temperature and $pCO_2$ was observed in the summer (p=0.013, supplementary material e)."

Line 240 "No temperature effect was observed...." But all response variables were higher in the summer than in the winter.

A: The effect of season has now been added (L. 246-247).

Line 248 "In the summer, the interactive effect of temperature and pCO2 increase was more complex, with a (change to) increase in Gd detected under high temperature conditions only."

A: The sentence has been changed: (L. 250-251) "In the winter, high $pCO_2$ increased net dissolutions rates, while in the summer $G_d$ increased under elevated temperature."

*Discussion*

The authors state that "ocean acidification and warming will strongly destabilize communities through both direct effects on species physiology and changes in the interaction strengths between coralline algae, fleshy algae, and grazers." Based on the assemblage data, I do not think that the effect is so negative, at least in the summer. There is a strong difference in the effect of the combination of CO2 and temperature in winter and summer. In summer, assemblages exposed to high temperature and pCO2 combined actually had similar to or even slightly higher light calcification rates than the ambient treatment. In winter, there was a decrease in light calcification compared to the ambient treatment, but the positive effect of temperature and the negative effect of pCO2 were weakened when the two were combined. I think it is important for the authors to point out that the combination of pCO2 and temperature often subdued the effects of each single factor, because it illustrates the point that experiments investigating only the effect of pCO2 or temperature may present more dramatic responses than when the two are combined, which represents a more realistic scenario.

A: The conclusion section has been reworded to consider this comment and those of Referee #1 and #3: "In conclusion, the community response to climate change does not appear to be only the result of individual species' metabolic responses, but also strongly depends on shifts in species interactions. In contrast with other studies, which evidenced larger impacts of the combination of increased $pCO_2$ and temperature than that of these factors alone (Reynaud et al., 2003; Anthony et al., 2008; Martin and Gattuso, 2009; Rodolfo-Metalpa et al., 2010), we showed here that the effects of $pCO_2$ and temperature on maerl bed communities were weakened when these factors were combined. Under the predicted business-as-usual conditions, epiphyte overgrowth may exacerbate the negative impact of climate change on underlying coralline algae. Here, we also demonstrated that climate change may affect grazer physiology, with major consequences on their ability to regulate epiphyte biomass. Climate change may also affect other components that we did not assess in the present study, such as

algal palatability and potential changes in grazer trophic behavior (Campbell et al., 2014; Duarte et al., 2015; Poore et al., 2013; Poore et al., 2016). Algal palatability to grazers may also be affected by predicted changes through shifts in the composition and the quantity of allelopathic compounds, as suggested by Del Monaco et al. (2017). In order to better understand the consequences of climate change on ecosystem functioning, further work should focus on the response of marine communities and consider more specifically shifts in species interactions, including changes in trophic interactions between algae and grazers."

*Technical Comments*

Line 129: insert "the" before CO2SYS

A: Done (L. 125)

It would be helpful to be able to identify statistically significant differences in the figures

A: (L. 197-198) "When 2-way AVNOVAs showed significant results, post hoc tests (Tukey honest significant difference, HSD) were performed to compare the four treatments." Results have been added on corresponding graphs. We have also added the direction of changes (2-way ANOVA) in tables 4, 5, 7 and 8 and interaction plots (in supplementary material) when a significant interaction between $pCO_2$ and temperature was detected.

---

## Author Comment (AC3) · 20 Sep 2017

We thank the referee and appreciate the thoughtful and constructive comments. We have fully considered the referee's comments and improved the manuscript accordingly. Answers to referee's comments are in red color.

The study by Legrand et al. assessed the metabolic responses of a range of species associated to maerl beds (incl coralline algae, grazers and epiphytic fleshy algae), as well as the metabolic responses of the maerl assemblage to changes in seawater carbonate chemistry and temperature across two climatic seasons. The authors found complex interactions among experimental factors and seasons on the species and community metabolism. The coralline algae exhibited responses which were expected under $CO_2$ perturbation experiments, but importantly, the study documented significant changes to grazers' metabolism and enhanced epiphytic algal biomass under $CO_2$ enrichment. Although ecological interactions were not directly assessed, changes in the metabolic responses of the experimental species are assumed to influence species interactions. Based on these results the authors were able to propose that ocean acidification and warming will have considerable impacts on the functioning of maerl beds.

I read this manuscript with great interest and believe the authors have done a comprehensive and thorough study. Most studies in the field of impacts of climate change on marine systems focus on responses of one or two species, generally within the same taxonomic group, and it is refreshing to see that this study took a step forward and assessed the impacts at the community level during two climatic seasons. Individual responses focussed on a range of response variables, incl chlorophyll (for the algae), net production, respiration, net calcification (light and dark), and excretion for the grazers. In combination with the assemblage's responses, this allowed the authors to discuss some potential ecological implication such as shifts in species composition, competition, carbon storage, etc. The Methods are generally well described and provide enough detail so that other researchers can repeat the experiments. Methods are appropriate for ocean acidification research.

**Main comments**

I have two main comments to the paper. First, seasonal effects on both the individual and assemblage responses were not fully explored or discussed in the m/s. One of the strengths of

this m/s is that it was conducted in two different climatic seasons, but how the strength of the responses varied between seasons was not clear. I would suggest that the authors include a section where this comment can be fully addressed.

Answer: We agree with this comment. A new paragraph has been added in the discussion to explore the seasonal effects on species and community metabolism. In the result section, the influence of season has been detailed for each species and the community. Further information was also added throughout the m/s to understand how the strength of the response varied between the two seasons tested.

(L. 262-267): "Assemblage exhibited a strong seasonal pattern for all metabolic parameters, which is consistent with the higher metabolism in the summer for most of the species incubated at the specific scale. This higher metabolism in the summer has already been evidenced in urchins (Brockington and Peck, 2001), gastropods (Davies, 1966; Innes and Houlihan, 1985; Martin et al., 2006b) and living maerl (Potin et al., 1990; Martin et al., 2006a) and is strongly related to changes in numerous environmental and biological variables, such as light intensity and photoperiod, temperature and nutrient or food availability (Godbold and Solan, 2013; Thomsen et al., 2013)."

The statistical analyses seem to be well executed, however, I would argue that because there were significant interactions between treatments (OA, temp, and season), there is a need to conduct further statistical analyses within treatment combinations, as in several instances, the main factor was significant, but in fact it was only significant for one or the other season, or under a particular treatment combination. For example, in line 213, "R was significantly reduced by the high temperature condition in the winter, whereas an increase in R was observed in the summer." This statement is fine, but is not actually supported by a statistical analysis as Table 3 only provides p values for the main effects. This issue is also evident 216-219. Underwood (1997; Experiments in Ecology: Their Logical Design and Interpretation Using Analysis of Variance, Cambridge University Press) provides information on this topic. These new analyses could be included as supplementary material.

A: According to the comments of Referee #1, we have modified statistical analyses. The seasonal effect has now been tested separately (using t-tests) in order to keep a balanced statistical design. The effect of temperature and $pCO_2$ was analyzed through 2-way ANOVA for each season separately. When an interactive effect of temperature and $pCO_2$ was observed, interaction plots were performed and provided in the supplementary material. (L. 197-198)

"When 2-way AVNOVAs showed significant results, post hoc tests (Tukey honest significant difference, HSD) were performed to compare the four treatments." (according to suggestions of Referee #2). Results have been shown in corresponding graphs.

There are some statements that are not supported by the experiments. Although the authors demonstrated changes in algal and grazer metabolisms, species interactions among those organisms were not examined experimentally. E.g. Line 251. "Our study demonstrates that the response of maerl bed communities to increased temperature and pCO2 conditions is a complex function of direct effects of climate variables on species physiology and shifts in species interactions". Reword this statement.

A: We agree with this comment and, as suggested by the reviewer, this sentence has been reworded: (L. 253-254) "The response of communities to increased temperature and pCO2 conditions is likely to be a complex function of direct effects of climate variables on species physiology and shifts in species interactions (Lord *et al.*, 2017)."

*Lord, J. P., Barry, J. P., and Graves, D.: Impact of climate change on direct and indirect species interactions, Marine Ecology Progress Series, 571, 1-11, 2017.*

**Minor comments**

Unclear why chla was measured on dead Lithothamnion. Provide a brief justification in section 3.3.

A: A sentence has been added in the "Material and Methods" section to justify chlorophyll a measurements: (L. 185-186) "In dead maerl, chlorophyll *a* content was measured in order to check for the presence of associated microflora and potential subsequent metabolism."

Line 90: In general avoid single-sentence paragraphs.

A: The sentence was grouped with the following paragraph

Line 237: ".. having positive effect". Was this effect significant?

A: More details have been provided due to the change in statistics recommended by referees: (L. 241-243) "Epiphyte biomass was not affected by increased temperature or $pCO_2$ in the winter (2-way ANOVA, p=0.95 and 0.67 respectively), while an interactive effect of temperature and $pCO_2$ was observed in the summer (p=0.013, supplementary material e)."

L260-280: This is a very long paragraph, try breaking it into two.

A: This paragraph has now been divided in 2 paragraphs in the revised manuscript.

L285-305: ´This is also a very long paragraph.

A: This paragraph has now been divided in 2 paragraphs in the revised manuscript.

L291: Ordonez et al. (Ordonez Alvarez et al. ´ 2014 Effects of ocean acidification on population dynamics and community structure of crustose coralline algae. Biological Bulletin 226, 255-268.) also found a failure in recruitment of tropical CCA and importantly documented shifts in species composition.

A: The reference has now been added in the revised manuscript. (L. 304)

Line 303: "However, the present findings do not support this idea, because a decline in Gl was observed under high pCO2 despite high". Short et al (2014) paper dealt with minute algal turfs which may have altered the thickness of the diffusive boundary layer on the coralline algae. The macroalgae investigated in the present study were much bigger and may interact in many different ways. It is perhaps very difficult to generalise the impacts of epiphytic algae on coralline algae given the diversity of algae in marine systems. Perhaps a line or two addressing this would be useful.

A: We agree with this comment. We have completed with the following sentences: (L. 314-325) "Conversely, other studies evidenced that the overgrowth of epiphytic fleshy algae may shade underlying coralline algae and reduce coralline net calcification rates (Garrabou and Ballesteros, 2000; Martin and Gattuso, 2009). The present findings support this idea, because a decline in assemblage $G_1$ was observed under high $pCO_2$ and high epiphyte biomass. […] Thus, overgrown maerl would be negatively affected by the direct effect of ocean acidification on calcification rates and indirect effects due to shifts in competition dynamics with fleshy epiphytic algae (Kuffner et al., 2008). However, the response of epiphytic algae is likely to be specie-specific and it appears difficult to generalize the impacts of epiphytic algae on coralline algae."

Pages 14-15: Grazing responses may also be altered by changes in seaweed allelopathic compounds, brought about by changes in composition, quantity, or in the magnitude/potency of the allelopathic interactions. A recent study showed that the potency of allelopathic interactions towards a tropical coral was intensified under ocean acidification conditions (Del

Monaco et al. 2017 Effects of ocean acidification on the potency of macroalgal allelopathy to a common coral. Scientific Reports 7, 41053). May be worth adding this potential mechanism as drivers of changes in species interactions in response to acidification and warming.

A: As suggested by the reviewer, we have now added this information: (L. 377-378) "Algal palatability to grazers may also be affected by predicted changes through shifts in the composition and the quantity of allelopathic compounds, as suggested by Del Monaco et al. (2017)."

---

## Editor Decision (ED1)

**BG-2017-255 Editor response to Revised Manuscript**

I would appreciate it if the points below could be addressed in a further revision. I do not believe the changes are especially onerous.

Reviewer #2

Line 216 The reviewer stated ' "Temperature had a positive effect on the Gl of living maerl. Conversely, Gl was significantly reduced under high pCO2..." The authors fail to mention that in the combined treatment, temperature alleviated the negative effect of pCO2. This is very important to the story. ', and the answer providd was as follows: A: The sentence has been revised due to the change in statistical design. (L. 218-219) "The Gl of living maerl was not significantly influenced by increased temperature and pCO2, regardless of the season" – To my mind this response is not clear. Do you mean that with the new statistical design there were no significant effects of either temperature or CO2, or that the combined treatment was not significantly different from the control? If the latter, then please at least make the suggestion mentioned by the reviewer that the individual effects of CO2 and temperature appeared to have canelled each other out.

Line 238: The reviewer asks you to state that 'temperature alone decreased epiphyte biomass in the summer', however the re-wording that you have made does not make this point clearly (you state 'while an interactive effect of temperature and pCO2 was observed in the summer'). I would prefer you to make a clear statement about the direction of this interaction and how the variables concerned were affected. If it is not possible to simply state that elevated temperature led to decreased epiphyte biomass in summer then please state why not (perhaps not supported by the new stats).

Discussion. The reviewer asks you to state more clearly that the combined effects of tempersture and CO2 may be less marked than the individual effects of those variables. In part of your re-worded section you state 'Under the predicted business-as-usual conditions, epiphyte overgrowth may exacerbate the negative impact of climate change on underlying coralline algae.' Please check this, as I think you may mean 'ameliorate' instead of 'exacerbate' (opposite meanings).

---

## Author Response (AR2)

**BG-2017-255 Editor response to Revised Manuscript**

We thank the editor for the constructive comments. We have considered the suggestions and improved the manuscript accordingly.

I would appreciate it if the points below could be addressed in a further revision. I do not believe the changes are especially onerous.

Reviewer #2

Line 216 The reviewer stated ' "Temperature had a positive effect on the Gl of living maerl. Conversely, Gl was significantly reduced under high pCO2..." The authors fail to mention that in the combined treatment, temperature alleviated the negative effect of pCO2. This is very important to the story. ', and the answer provided was as follows: A: The sentence has been revised due to the change in statistical design. (L. 218-219) "The Gl of living maerl was not significantly influenced by increased temperature and pCO2, regardless of the season" – To my mind this response is not clear. Do you mean that with the new statistical design there were no significant effects of either temperature or CO2, or that the combined treatment was not significantly different from the control? If the latter, then please at least make the suggestion mentioned by the reviewer that the individual effects of CO2 and temperature appeared to have cancelled each other out.

Answer: According to the now statistical design, no significant effect of temperature and $pCO_2$ was detected on $G_l$ (see table below). We have kept the statement: "The $G_l$ of living maerl was not significantly influenced by increased temperature and pCO2, regardless of the season (Table 5, Fig. 2d)" (lines 219-220).

| | | Light calcification $G_l$ | |
|---|---|---|---|
| | **WINTER** | F | p |
| | T | 3.6 | 0.08 |
| **LIVING** *L. corallioides* | $pCO_2$ | 3.2 | 0.09 |
| | $pCO_2$ x T | 0.8 | 0.39 |
| | **SUMMER** | | |
| | *T* | *3.3* | *0.07* |
| | *$pCO_2$* | *3.6* | *0.06* |
| | *$pCO_2$ x T* | *0.2* | *0.65* |

Line 238: The reviewer asks you to state that 'temperature alone decreased epiphyte biomass in the summer', however the re-wording that you have made does not make this point clearly (you state 'while an interactive effect of temperature and pCO2 was observed in the summer'). I would prefer you to make a clear statement about the direction of this interaction and how the variables concerned were affected. If it is not possible to simply state that elevated temperature led to decreased epiphyte biomass in summer then please state why not (perhaps not supported by the new stats).

A: We have reworded our statement to consider the direction of the interaction for each variable: "In the summer, increased temperature reduced epiphyte biomass under ambient $pCO_2$ and stimulated epiphyte biomass under elevated $pCO_2$ (2-way ANOVA, p=0.013, supplementary material e)." (lines 242-243).

Discussion. The reviewer asks you to state more clearly that the combined effects of temperature and CO2 may be less marked than the individual effects of those variables. In part of your re-worded section you state 'Under the predicted business-as-usual conditions, epiphyte overgrowth may exacerbate the negative impact of climate change on underlying coralline algae.' Please check this, as I think you may mean 'ameliorate' instead of 'exacerbate' (opposite meanings).

A: In the conclusion, we stated that the combined effects of temperature and $CO_2$ ameliorate the response of assemblages: "In contrast with other studies, […] 
[revised manuscript text omitted]